# Impact of global short-term landscape fire sourced PM$_{2.5}$ exposure on child cause-specific morbidity: a study in multiple countries and territories

Shuang Zhou[1], Yiwen Zhang [1], Zhengyu Yang [1], Rongbin Xu [1,2], Wenzhong Huang [1], Yao Wu [1], Zhihu Xu [1], Yuan Gao[1], Yanming Liu[1], Wenhua Yu [1], Pei Yu [1], Gongbo Chen[1], Ke Ju[1], Tingting Ye [1], Bo Wen [1], Yuxi Zhang [3], Michael Abramson [1], Lidia Morawska [4], Fay H. Johnston [5], Simon Hales [6], Micheline S. Z. S. Coelho [7], Yue Leon Guo [8], Jane Heyworth [9], Wissanupong Kliengchuay[10], Luke Knibbs[11], Eric Lavigne [12], Guy Marks[13], Patricia Matus [14], Geoffrey Morgan [11], Paulo H. N. Sadiva[7], Kraichat Tantrakarnapa [10], Yuming Guo [1] ✉ & Shanshan Li [1] ✉

Children are particularly vulnerable to landscape fire sourced fine particulate matter (LFS PM$_{2.5}$), yet evidence on its health effects remains limited. Here we show that short-term exposure to LFS PM$_{2.5}$ is associated with increased hospital admissions for multiple diseases in children and adolescents. We analysed daily hospital admission data from 1012 communities in seven countries/territories, linked to a high-resolution LFS PM$_{2.5}$ dataset. Each 10 μg/m$^3$ increase in LFS PM$_{2.5}$ was associated with elevated risks for all-cause (1.1%), respiratory (1.9%), infectious (1.5%), cardiovascular (2.9%), neurological (2.8%), diabetes (3.7%), cancer (1.5%), and digestive (0.8%) hospital admissions. Risks for respiratory, infectious, and neurological conditions increased even at low exposure, while others rose only above 15-20 μg/m$^3$. Children aged 5-9 years and those in lower socioeconomic areas were especially affected. These findings highlight the health burden of LFS PM$_{2.5}$ in young people and the urgent need to reduce exposure and protect vulnerable populations.

Landscape fires occur in natural (e.g., forests, grassland) and cultural (e.g., farmlands) landscapes, and include both planned and unplanned fires such as anthropogenic fires and wildfires[1]. Landscape fires can directly cause injuries and indirectly affect human health through air pollution, ecological destruction, and infrastructure disruption[2]. Compared to the direct effects of flames, landscape fire-sourced (LFS) air pollution can travel thousands of kilometers, affecting larger populations and posing a greater public health risk[3]. Globally, more than 2 billion people were exposed to LFS air pollution each year during 2010–2019[1]. Among these pollutants, LFS fine particulate matter ≤2.5 μm in diameter (PM$_{2.5}$) is particularly concerning due to its severe and widespread health impacts[4,5]. LFS PM$_{2.5}$ exposure shows substantial geographic variation. High levels are typically observed in fire-prone regions such as South America, Southeast Asia, and northern Australia, while non-fire-prone areas may also experience considerable exposure due to long-range smoke transport[1]. During transport, LFS PM$_{2.5}$ can become more toxic through chemical transformation, potentially increasing its health impacts[1,3].

Inhaled LFS PM$_{2.5}$ can deposit deep into the lungs to trigger respiratory inflammation and oxidative stress, leading to respiratory

damage and increased susceptibility to infections[6]. Its toxic components, such as ultrafine particles and heavy metals, can enter the bloodstream to cause systemic inflammation, oxidative stress, and immune dysfunction[7]. These biological pathways suggest that LFS $PM_{2.5}$ may increase risks for multiple organ systems, as widely documented in adults[4,5].

Children exhibit greater vulnerability to LFS $PM_{2.5}$, possibly due to behaviors such as spending more time outdoors and the immaturity of their respiratory and immune systems[8]. Globally, LFS $PM_{2.5}$ contributed to an estimated 12.9 million deaths in children and adolescents during 2000-2014, accounting for 9.1% of the total[9]. Previous studies examining short-term LFS $PM_{2.5}$ exposure in children have primarily focused on respiratory diseases[8,10]. Some studies conducted in Mozambique[11], Brazil[12], California[13], and Washington State[14] reported positive associations, while others in Colorado[15] and San Diego[16] found no significant effects. A recent review has linked LFS $PM_{2.5}$ to an increased risk of upper respiratory infections, while the evidence remains inconclusive for other health effects, such as asthma and bronchitis[10]. However, research on the broader health effects of LFS $PM_{2.5}$ in children is limited, particularly regarding its potential associations with infectious, cardiovascular, neurological, metabolic, digestive, and renal diseases[8,10].

Most previous studies were conducted in single locations[11–17], limiting the ability to compare and synthesize findings across different regions. Although some research spanned multiple countries, they often focused on specific age groups, such as 0–5 years or specific health outcomes, such as respiratory diseases[9,18]. This narrow focus might limit the understanding of age-specific vulnerabilities and the broader spectrum of morbidity related to LFS $PM_{2.5}$. To our knowledge, no previous study has comprehensively evaluated the short-term impacts of LFS $PM_{2.5}$ on a wide range of cause-specific morbidity outcomes in children and adolescents across different regions and populations of the world.

Here, we show that short-term exposure to LFS $PM_{2.5}$ exposure is associated with increased hospital admissions for a broad range of diseases in children and adolescents, including respiratory, infectious, cardiovascular, neurological, metabolic, and digestive conditions. By analyzing data from 1012 communities across seven countries/territories, covering both fire-prone regions (Brazil, Chile, Australia, Canada, and Thailand) and non-fire-prone regions (New Zealand and Taiwan), we provide a comprehensive understanding of the morbidity risks from LFS $PM_{2.5}$ to children and inform more effective prevention and mitigation policies.

## Results

### Summary of LFS $PM_{2.5}$ levels and hospital admissions

Communities in Brazil, Chile, and Thailand experienced higher LFS $PM_{2.5}$ levels, compared with other regions (Fig. 1). The median concentration of LFS $PM_{2.5}$ was 1.2 μg/m³ [interquartile range (IQR, 25th–75th): 0.4–3.4 μg/m³], with substantial regional variation (Table S1). Among the study locations, Chile recorded the highest LFS $PM_{2.5}$ exposure, with a median concentration of 9.5 μg/m³ (IQR: 5.1–15.5 μg/m³), followed by Brazil (2.2 μg/m³, IQR: 0.9–5.7 μg/m³) and Thailand (1.6 μg/m³, IQR: 0.6–7.7 μg/m³). Moderate levels were observed in Australia (1.1 μg/m³, IQR: 0.5–2.5 μg/m³) and Taiwan (1.1 μg/m³, IQR: 0.5–1.9 μg/m³), while New Zealand (0.7 μg/m³, IQR: 0.3–1.6 μg/m³) had slightly lower levels. The lowest LFS $PM_{2.5}$ concentrations were recorded in Canada (0.3 μg/m³, IQR: 0.1–1.0 μg/m³). During 2000 to 2019, 67.9 million all-cause [31.4 million (46.3%) girls] hospital admissions among children and adolescents aged 0–19 years were recorded (Table S2).

### Association of LFS $PM_{2.5}$ and hospital admission

**Lag-response association.** Figure 2 shows the lag patterns for all-cause and cause-specific hospital admission risks in children and

adolescents associated with LFS $PM_{2.5}$ during lag 0–14 days. The highest risks occurred on the current day (lag 0) of LFS $PM_{2.5}$ exposure and decreased afterwards, with significantly positive associations on lag 0 and lag 1 days. Most lag-response associations were non-significant from lag 2 days, while the association for respiratory hospital admissions was still significantly positive on lag 2 days, and neurological hospital admissions was significantly positive on lag 2 and lag 3 days. Therefore, we estimated cumulative RR during the first two days (lag 0–1) in the following analysis.

**Exposure–response association.** The cumulative exposure–response relationship between LFS $PM_{2.5}$ during lag 0–1 days and hospital admission risk was monotonically increasing and approximately linear (Fig. 3). The cumulative RRs for all-cause, respiratory, and infectious hospital admissions increased significantly as $PM_{2.5}$ concentrations rose from 0 μg/m³, with a steady rise as concentrations increased. The RRs for neurological hospital admissions began increasing from 0 μg/m³, though the rise was more gradual between 0 and 10 μg/m³, before becoming steeper at higher concentrations. The exposure–response curves for cardiovascular, diabetes, cancer, digestive, and renal diseases showed no increased risks at lower concentrations, followed by a sharper increase in risk at higher levels from 10 to 25 μg/m³.

**Cumulative relative risks.** After adjusting for multiple comparisons using false discovery rate (FDR) method, a 10 μg/m³ increase in LFS $PM_{2.5}$ was associated with increased risks of all-cause [RR: 1.011 (95%CI: 1.009, 1.012)], respiratory [1.019 (1.015, 1.022)], infectious [1.015 (1.010, 1.019)], cardiovascular [1.029 (1.017, 1.041)], neurological [1.028 (1.016, 1.041)], diabetes [1.037 (1.018, 1.057)], cancer [1.015 (1.007, 1.023)], and digestive [1.008 (1.005, 1.012)] hospital admissions. The association with renal disease admissions [1.010 (1.000, 1.020)] was no longer statistically significant after FDR adjustment (Table 1).

Geographical heterogeneity existed in the conditions sensitive to LFS $PM_{2.5}$ across different countries/territories after FDR adjustment (Table 1). In Australia, an increased risk was only observed for renal hospital admissions [1.060 (1.006, 1.118)]. In Brazil, increased risks were observed for all-cause, respiratory, infectious, neurological, diabetes, and cancer hospital admissions, with the highest RR for diabetes [1.047 (1.011, 1.086)]. In Canada, increased risks were observed for infectious [1.071 (1.035, 1.108)], cardiovascular [1.170 (1.070, 1.279)], diabetes [1.091 (1.017, 1.169)], and cancer [1.098 (1.025, 1.176)] hospital admissions. Chile showed increased risks in all-cause, respiratory, cardiovascular, neurological, and digestive hospital admissions, with the highest RR for cardiovascular [1.027 (1.011, 1.044)]. In New Zealand, increased risks were observed for all causes [1.022 (1.010, 1.035)] and respiratory [1.045 (1.018, 1.072)] hospital admissions. Thailand experienced increased risks in all-cause, respiratory, infectious, neurological, and digestive hospital admissions. In Taiwan, increased risks were observed for all causes and respiratory hospital admissions. Inconsistency of effect estimates was moderate or strong ($I^2 \geq 40\%$) within most countries/territories.

**Modification effects.** Stratified analyses by age and sex after FDR adjustment are shown in Fig. 4 and Table 1. For all-cause and infectious hospital admissions, higher risks were observed among children aged 5–9 years than those aged 0–4 or 10–19 years. Compared with children aged 5–9 years, those aged 10–19 years showed a lower risk for renal hospital admissions. Generally, the associations between LFS $PM_{2.5}$ and all-cause and cause-specific hospital admissions were similar between females and males.

Similar significant effect modification by local GDP per capita and country income class was observed after FDR adjustment (Table 1, Figs. S1 and S2). Overall, children in the communities with a low or medium GDP per capita and upper-middle income countries showed higher risks of all-cause and digestive hospital admissions than those in

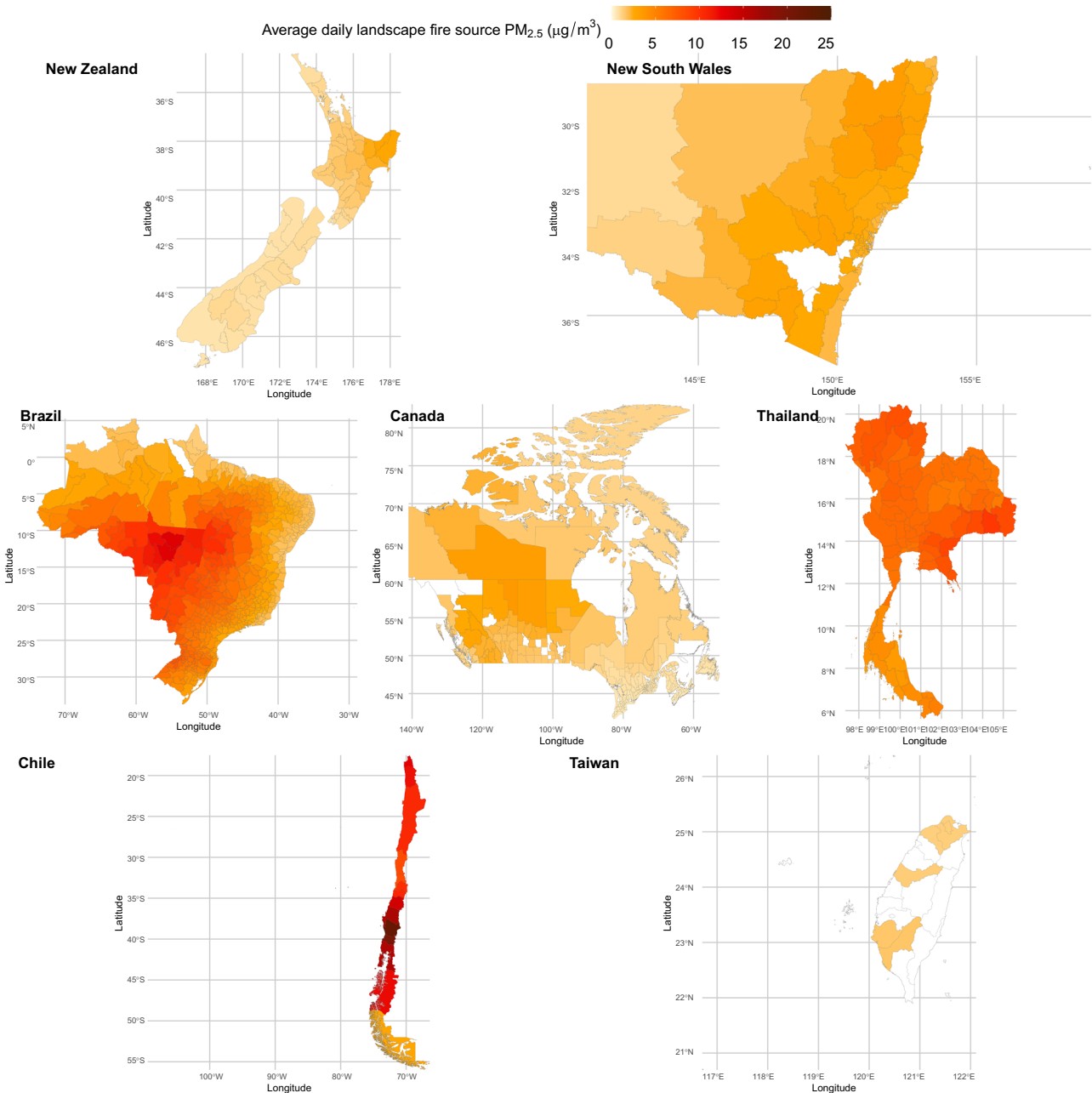

**Fig. 1 | The spatial distributions of average concentrations of daily LFS PM$_{2.5}$ in 1012 communities during 2000–2019.** The map was created in R (version 4.1.1) using shapefiles downloaded from OpenStreetMap via the QuickOSM plugin in QGIS (version 3.32.3). OpenStreetMap data are licensed under the Open Database License (ODbL) (https://www.openstreetmap.org/copyright).

the communities with a high GDP per capita and high-income countries.

### Result of sensitivity analyses
The results remained robust after changing the knots for LFS PM$_{2.5}$ and the df for meteorological variables or time trend (Figs. S3 and S4).

## Discussion
To our knowledge, this is the most comprehensive multi-country/territory study to data exploring the associations between LFS PM$_{2.5}$ and child morbidity. We observed that LFS PM$_{2.5}$ was associated with increased risks of all-cause, respiratory, infectious, cardiovascular, neurological, diabetes, cancer, and digestive hospital admissions among children. The exposure–response relationships varied across different diseases. Risks for all-cause, respiratory, infectious, and

neurological admissions increased from 0 µg/m$^3$, though the increase for neurological admissions was more gradual at lower concentrations. In contrast, risks for other diseases were near zero at low concentrations and rose at higher levels. Geographical heterogeneity existed, and significant effect modifications by age, local GDP, and country income class were observed. Children aged 5–9 years showed higher risks for all-cause and infectious hospital admissions. Children in regions with lower local GDP and country income level showed higher risks of all-cause hospital admissions.

The relationship between LFS PM$_{2.5}$ and respiratory outcomes aligned with findings from previous studies[11–14,18]. Specifically, a study from southern Mozambique reported that each 10 µg/m$^3$ increment in cumulative LFS PM$_{2.5}$ for lag 0–1 days was associated with a 15.6% increment in all-cause and a 27.0% increment in respiratory hospital visits[11]. Another study from Brazil reported a 2.3% increase in all-cause

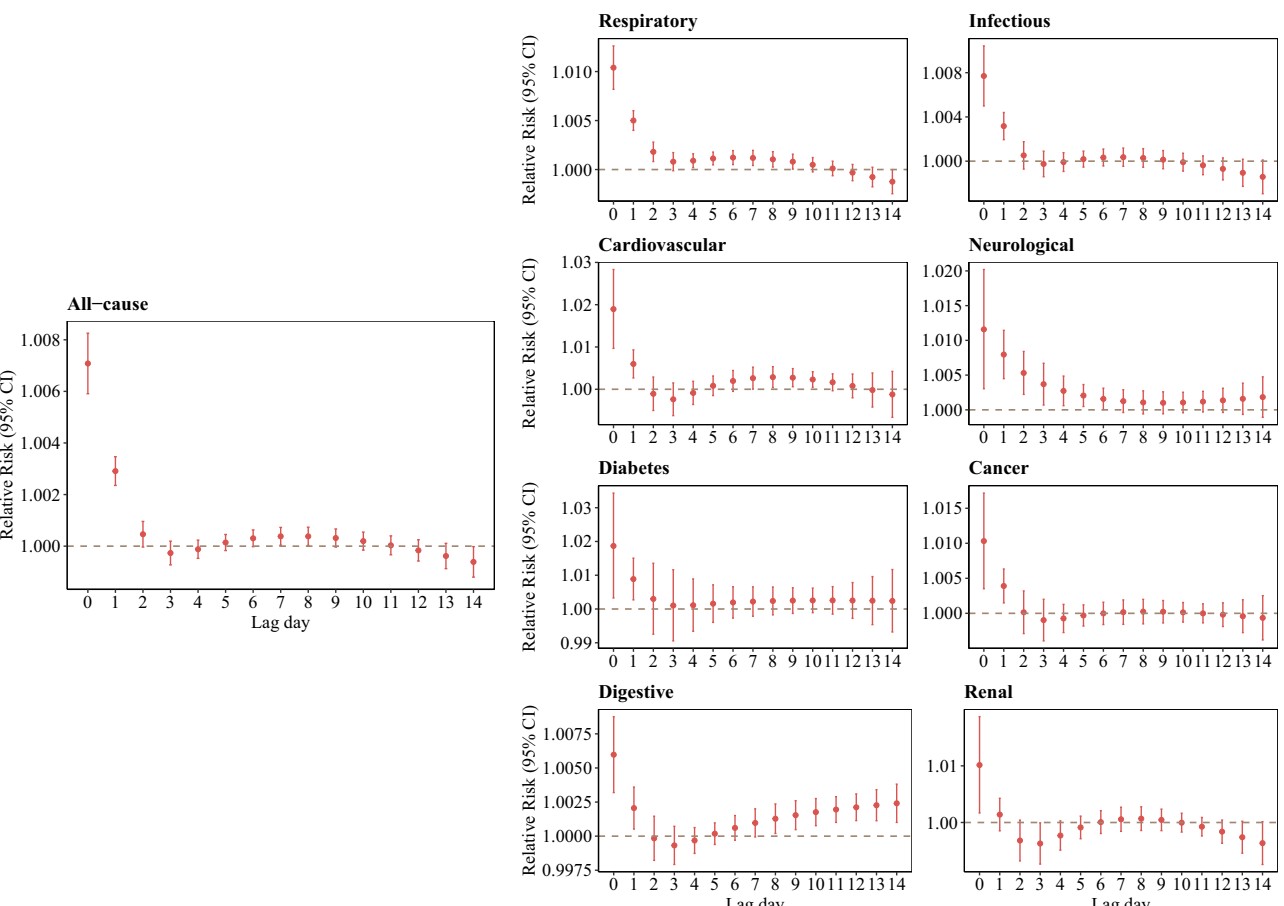

**Fig. 2 | Relative risks of all-cause and cause-specific hospital admissions in children and adolescents aged 0–19 years associated with 10 µg/m³ increase in LFS PM$_{2.5}$ during lag 0–14 days in 1012 communities.** Data are presented as estimated relative risks (RRs) with 95% confidence intervals (CIs). Central points represent the estimated RRs, and error bars indicate the corresponding 95% CIs. Estimates are derived from a two-stage time-series analysis.

hospital admissions for children aged 5–9 and a 4.9% increase for children aged 0-4[12]. Similarly, studies in California[13] and Washington State[14] also found increased risks of respiratory hospital admissions in children associated with LFS PM$_{2.5}$ exposure for lag 0–1 days (RR: 1.026) and lag 0 day (RR: 1.069), respectively. However, studies in Colorado[15] and San Diego[16] did not observe significant associations. Cardiovascular disease is another major health outcome linked to LFS PM$_{2.5}$ exposure, but most research focused on adults[19]. Our study also identified a significant association between LFS PM$_{2.5}$ exposure and higher risk of cardiovascular hospital admissions in children, whereas previous studies in California, Washington, and Colorado did not report significant associations[13–15]. These discrepancies may result from variations in LFS PM$_{2.5}$ concentrations, exposure assessment methods, study populations, and differences in healthcare infrastructure and health-seeking behaviors. Additionally, our study identified associations between LFS PM$_{2.5}$ with infectious, neurological, diabetes, cancer, and digestive hospital admissions, suggesting a broader impact of LFS PM$_{2.5}$ on child health than previously recognized and reinforcing the need for further research to validate these findings.

The observed associations between LFS PM$_{2.5}$ exposure and increased hospital admissions can be explained by its biological mechanisms. Upon inhalation, LFS PM$_{2.5}$ deposits deep into the respiratory tract and triggers inflammation and oxidative stress, which can lead to lung damage and increased susceptibility to infections[20–22]. Some toxic components (e.g., ultrafine particles and heavy metals) can enter the bloodstream and induce systemic inflammation that may led to cardiovascular[23], metabolic, and other diseases[24]. Beyond the lungs,

LFS PM$_{2.5}$ can reach the brain via the olfactory nerve or bypassing the blood-brain barrier, which will lead to neuroinflammation and increase the risks of neurological disorders[25]. LFS PM$_{2.5}$ can also be ingested through contaminated food and water or transported to the gut via mucociliary clearance. Once in the digestive system, it may disrupt gut microbiota, increase intestinal permeability, and promote inflammation, resulting in digestive diseases[26]. Additionally, fire smoke exposure weakens immune function, increasing susceptibility to respiratory and systemic infections[27]. These mechanisms provide a biological explanation for the widespread health impacts observed in our study.

The lag patterns observed in our study align with these mechanistic pathways. The observed largest effects at lag 0–1 days suggest an acute inflammation and oxidative stress response to LFS PM$_{2.5}$ exposure, consistent with previous studies[11–14]. The attenuation of these associations at longer lags may result from both the rapid biological response to acute LFS PM$_{2.5}$ exposure and the dispersion of LFS PM$_{2.5}$ over time, leading to lower exposure levels and reduced direct impact on health[3]. However, digestive diseases showed a secondary rise in risk around lag 7, potentially due to delayed effects from mucociliary clearance and ingestion of fire-related pollutants[26]. This prolonged effect highlights the need for clinicians to monitor for delayed-onset symptoms and for healthcare systems to allocate resources accordingly to ensure adequate medical support beyond the immediate exposure period.

The exposure–response relationships observed further supported these mechanistic pathways. Risks for all-cause, respiratory, and infectious hospital admissions increased steadily from low levels of LFS PM$_{2.5}$ exposure, indicating that even minimal exposure exacerbated

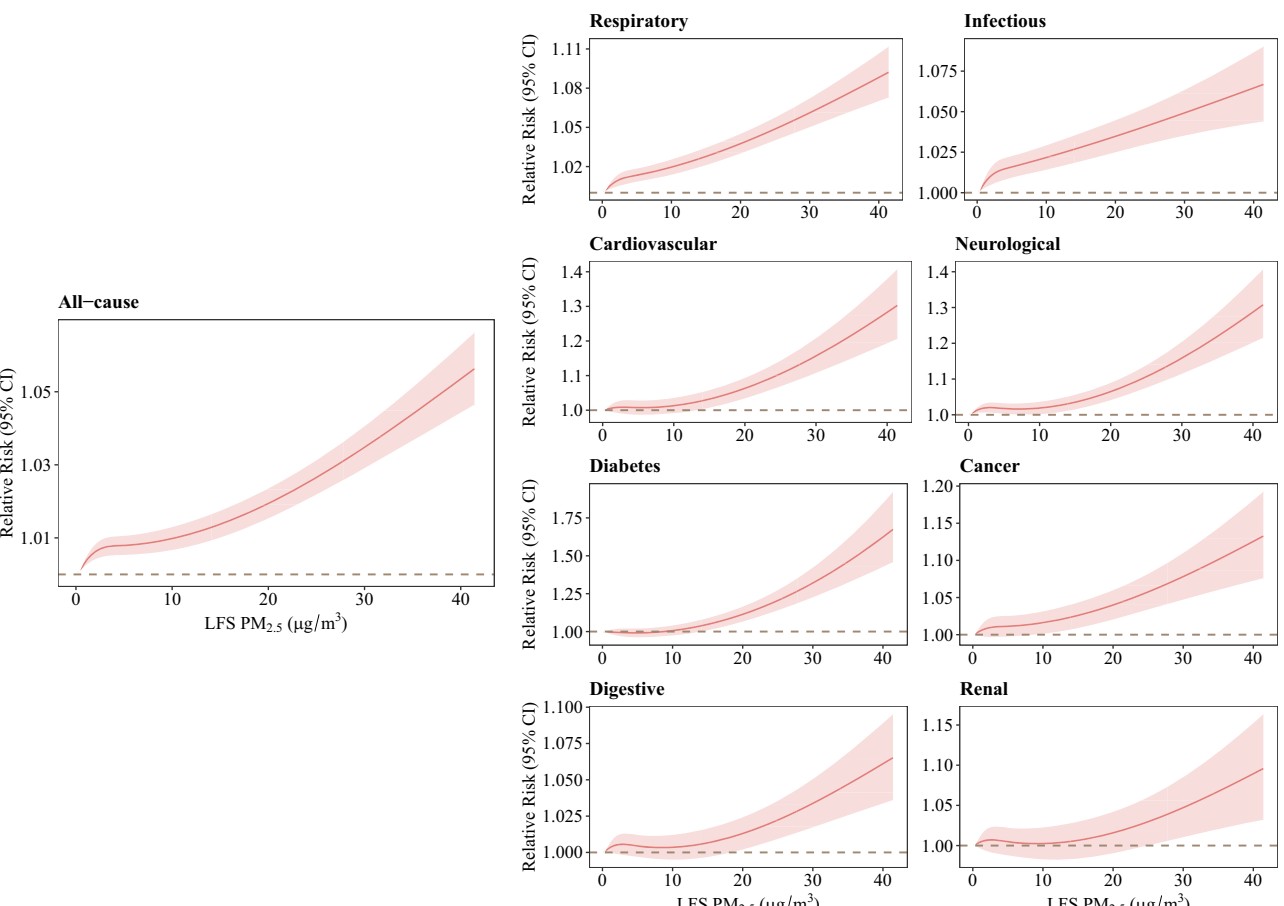

**Fig. 3 | The cumulative exposure–response curves for the associations between LFS PM₂.₅ during lag 0–1 days and all-cause and cause-specific hospital admissions in children and adolescents aged 0–19 years in 1012 communities.** Data are presented as estimated relative risks (central solid lines) with 95% confidence intervals (shaded areas). Curves were estimated using a two-stage time-series model with a natural cubic spline and two knots placed at the 25th and 75th percentiles of the mean LFS PM₂.₅ distribution.

respiratory inflammation and immune responses[4]. Neurological diseases showed a more gradual increase in risk at very low exposure levels, which may reflect both the ability of LFS PM₂.₅ to reach the brain even at low exposure levels and the slower progression of neuroinflammation compared to acute respiratory and immune responses[28,29]. In contrast, cardiovascular (mainly peripheral vascular diseases in children of this study), diabetes (mainly Type I diabetes), cancer (mainly leukemia), digestive (mainly acute pancreatitis), and renal (mainly renal tubulointerstitial diseases) risks increased only at higher concentrations, around 15–20 μg/m³, suggesting more sustained or higher exposure is needed to trigger systemic effects through oxidative stress and inflammation. Especially for pediatric patients with pre-existing diabetes or cancer, short-term LFS PM₂.₅ may compromise their immune function and exacerbate their pollution-induced complications rather than triggering new-onset cases[24,30]. Although cardiovascular hospital admissions are relatively uncommon in children, short-term exposure to LFS PM₂.₅ may worsen pre-existing conditions such as congenital heart disease or myocarditis and lead to hospitalization. Further research is needed to understand these exposure–response relationships, particularly at higher LFS PM₂.₅ concentrations.

We observed geographical heterogeneity in hospital admissions linked to LFS PM₂.₅ across different countries/territories, including both fire-prone and non-fire-prone regions. These differences may stem from variations in vegetation type[31], LFS events pattern[32], population demographics, environmental policies, healthcare infrastructure, and help-seeking behaviors[33]. Interestingly, those non-fire-prone regions like New Zealand and Taiwan showed relatively higher risks for all-cause and respiratory hospital admission associated with LFS PM₂.₅ than those fire-prone regions. This may be due to the chemical transformation of LFS during long-range transport, which can increase oxidative and proinflammatory components of LFS PM₂.₅[34]. Additionally, these regions experience LFS events less frequently, thus they may lack adequate response systems, leading to higher hospital admissions when exposure occurs[35]. These findings highlight the need for region-specific public health interventions tailored to local environmental conditions and vulnerabilities.

Socioeconomic status (SES) also influences the health impacts of LFS PM₂.₅, with lower-SES regions showing higher risks of all-cause and digestive-related hospital admissions. Several factors may explain this disparity. First, lower-SES regions often experience higher LFS PM₂.₅ concentrations, while people in these areas may have fewer resources to adopt protective measures such as indoor air filtration systems[1]. Second, children in lower-SES areas tend to have poorer baseline health status and limited healthcare programs (e.g., routine health check-ups, vaccinations, and comprehensive public health education programs)[36], making them more vulnerable to LFS PM₂.₅. Third, inadequate healthcare access can lead to delayed treatment, worsening disease severity. Fourth, limited policy implementation addressing wildfire events and air pollution in lower-SES areas further exacerbates health risks[37]. This finding underscores the need for targeted interventions such as improving indoor air quality through air filtration systems, increasing public awareness through education programs, and strengthening policies to mitigate the disproportionate burden of LFS PM₂.₅ in lower-SES regions.

**Table 1 | The cumulative relative risks (and 95% Confidence Intervals) of all-cause and cause-specific hospital admissions in children and adolescents aged 0–19 years associated with a 10 μg/m³ increase in LFS PM$_{2.5}$ during lag 0–1 day**

| Subgroups | Relative Risk (95% CI) | | | | | | | | |
|---|---|---|---|---|---|---|---|---|---|
| | All-cause | Respiratory | Infectious | Cardiovascular | Neurological | Diabetes | Cancer | Digestive | Renal |
| Overall | 1.011 (1.009, 1.012)[a,b] | 1.019 (1.015, 1.022)[a,b] | 1.015 (1.010, 1.019)[a,b] | 1.029 (1.017, 1.041)[a,b] | 1.028 (1.016, 1.041)[a,b] | 1.037 (1.018, 1.057)[a,b] | 1.015 (1.007, 1.023)[a,b] | 1.008 (1.005, 1.012)[a,b] | 1.010 (1.000, 1.020)[a] |
| **Country/Territory** | | | | | | | | | |
| Australia | 0.999 (0.995, 1.003) | 1.010 (0.998, 1.023) | 1.003 (0.985, 1.022) | 1.030 (0.985, 1.077) | 1.017 (0.996, 1.038) | 1.044 (0.980, 1.111) | 1.024 (0.983, 1.067) | 0.987 (0.975, 1.000)[a] | 1.060 (1.006, 1.118)[a,b] |
| Brazil | 1.011 (1.008, 1.013)[a,b] | 1.020 (1.015, 1.025)[a,b] | 1.011 (1.004, 1.018)[a,b] | 1.016 (0.996, 1.036) | 1.033 (1.011, 1.055)[a,b] | 1.047 (1.011, 1.086)[a,b] | 1.023 (1.010, 1.037)[a,b] | 1.003 (0.995, 1.011) | 1.003 (0.988, 1.018) |
| Canada | 1.001 (0.995, 1.007) | 1.015 (0.992, 1.039) | 1.071 (1.035, 1.108)[a,b] | 1.170 (1.070, 1.279)[a,b] | 1.031 (0.982, 1.082) | 1.091 (1.017, 1.169)[a,b] | 1.098 (1.025, 1.176)[a,b] | 1.002 (0.984, 1.019) | 1.054 (0.979, 1.136) |
| Chile | 1.007 (1.003, 1.010)[a,b] | 1.010 (1.003, 1.017)[a,b] | 1.003 (0.997, 1.009) | 1.027 (1.011, 1.044)[a,b] | 1.015 (1.006, 1.024)[a,b] | 1.020 (0.996, 1.044) | 1.002 (0.993, 1.011) | 1.007 (1.003, 1.012)[a,b] | 1.003 (0.987, 1.018) |
| New Zealand | 1.022 (1.010, 1.035)[a,b] | 1.045 (1.018, 1.072)[a,b] | 0.998 (0.949, 1.050) | 0.961 (0.793, 1.165) | 1.088 (1.005, 1.178)[a] | 0.997 (0.806, 1.232) | 0.929 (0.828, 1.042) | 1.033 (0.993, 1.076) | 0.851 (0.724, 1.000)[a] |
| Thailand | 1.019 (1.015, 1.023)[a,b] | 1.018 (1.013, 1.024)[a,b] | 1.022 (1.016, 1.027)[a,b] | 1.031 (0.979, 1.086) | 1.027 (1.003, 1.050)[a,b] | 0.963 (0.693, 1.338) | 1.038 (1.001, 1.076)[a] | 1.031 (1.019, 1.043)[a,b] | 1.030 (0.992, 1.070) |
| Taiwan | 1.041 (1.025, 1.057)[a,b] | 1.088 (1.057, 1.120)[a,b] | 0.987 (0.920, 1.059) | | | | | 1.097 (1.006, 1.196)[a] | |
| **Sex** | | | | | | | | | |
| Female | 1.010 (1.008, 1.013)[a,b] | 1.020 (1.015, 1.024)[a,b] | 1.016 (1.011, 1.021)[a,b] | 1.027 (1.002, 1.052)[a,b] | 1.043 (1.025, 1.061)[a,b] | 1.042 (0.981, 1.106) | 1.026 (1.015, 1.037)[a,b] | 1.007 (1.003, 1.012)[a,b] | 1.023 (1.007, 1.039)[a,b] |
| Male | 1.011 (1.009, 1.013)[a,b] | 1.018 (1.014, 1.022)[a,b] | 1.013 (1.009, 1.018)[a,b] | 1.036 (1.022, 1.050)[a,b] | 1.032 (1.015, 1.049)[a,b] | 1.041 (1.014, 1.069)[a,b] | 1.008 (0.996, 1.019) | 1.011 (1.006, 1.015)[a,b] | 1.014 (0.996, 1.032) |
| **Age** | | | | | | | | | |
| 0–4 | 1.010 (1.008, 1.012)[a,b] | 1.016 (1.013, 1.020)[a,b] | 1.013 (1.008, 1.017)[a,b] | 1.019 (0.981, 1.059) | 1.018 (1.007, 1.029)[a,b] | 0.962 (0.772, 1.200) | 1.014 (0.998, 1.030) | 1.014 (1.007, 1.021)[a,b] | 1.034 (1.013, 1.055)[a,b] |
| 5–9 | 1.016 (1.013, 1.019)[a,b] | 1.024 (1.018, 1.030)[a,b] | 1.025 (1.017, 1.033)[a,b] | 1.041 (0.985, 1.099) | 1.031 (1.015, 1.047)[a,b] | 0.909 (0.668, 1.236) | 1.024 (1.005, 1.043)[a,b] | 0.997 (0.984, 1.010) | 1.031 (1.009, 1.053)[a,b] |
| 10–19 | 1.008 (1.006, 1.011)[a,b] | 1.024 (1.018, 1.031)[a,b] | 1.008 (1.000, 1.016) | 1.031 (1.017, 1.044)[a,b] | 1.047 (1.020, 1.074)[a,b] | 1.027 (1.007, 1.047)[a,b] | 1.038 (1.017, 1.061)[a,b] | 1.008 (1.003, 1.012)[a,b] | 0.995 (0.981, 1.008) |
| **Country income class** | | | | | | | | | |
| Upper-middle | 1.013 (1.011, 1.015)[a,b] | 1.019 (1.015, 1.022)[a,b] | 1.014 (1.010, 1.019)[a,b] | 1.023 (1.011, 1.034)[a,b] | 1.030 (1.015, 1.046)[a,b] | 1.030 (1.007, 1.053)[a,b] | 1.011 (1.003, 1.019)[a,b] | 1.010 (1.006, 1.015)[a,b] | 1.007 (0.997, 1.017) |
| High | 1.003 (1.000, 1.007) | 1.020 (1.009, 1.030)[a,b] | 1.016 (1.001, 1.031)[a] | 1.061 (1.016, 1.107)[a,b] | 1.023 (1.005, 1.041)[a,b] | 1.058 (1.013, 1.105)[a,b] | 1.032 (0.998, 1.067) | 0.996 (0.986, 1.006) | 1.051 (1.009, 1.093)[a,b] |
| **GDP** | | | | | | | | | |
| Low | 1.014 (1.011, 1.017)[a,b] | 1.022 (1.017, 1.028)[a,b] | 1.015 (1.009, 1.021)[a,b] | 1.010 (0.983, 1.036) | 1.025 (1.004, 1.047)[a,b] | 1.065 (1.003, 1.129)[a] | 1.025 (1.005, 1.045)[a,b] | 1.017 (1.008, 1.026)[a,b] | 1.016 (0.997, 1.035) |
| Middle | 1.011 (1.009, 1.014)[a,b] | 1.016 (1.011, 1.021)[a,b] | 1.013 (1.007, 1.019)[a,b] | 1.028 (1.015, 1.042)[a,b] | 1.036 (1.012, 1.061)[a,b] | 1.024 (1.004, 1.045)[a,b] | 1.007 (0.999, 1.015) | 1.007 (1.003, 1.012)[a,b] | 1.002 (0.990, 1.014) |
| High | 1.002 (0.999, 1.006) | 1.017 (1.007, 1.027)[a,b] | 1.014 (0.998, 1.030) | 1.054 (1.008, 1.102)[a,b] | 1.024 (1.006, 1.043)[a,b] | 1.063 (1.017, 1.112)[a,b] | 1.028 (0.994, 1.064) | 0.995 (0.985, 1.006) | 1.047 (1.004, 1.093)[a,b] |

[a] P_unadjusted < 0.05, P values are not adjusted for multiple comparisons.
[b] P_adjusted < 0.05, P values are adjusted for multiple comparisons using the FDR method. The adjustment was implemented using the Benjamini–Hochberg (BH) method via the p.adjust function (method = "fdr") in R. All statistical tests were two-sided.

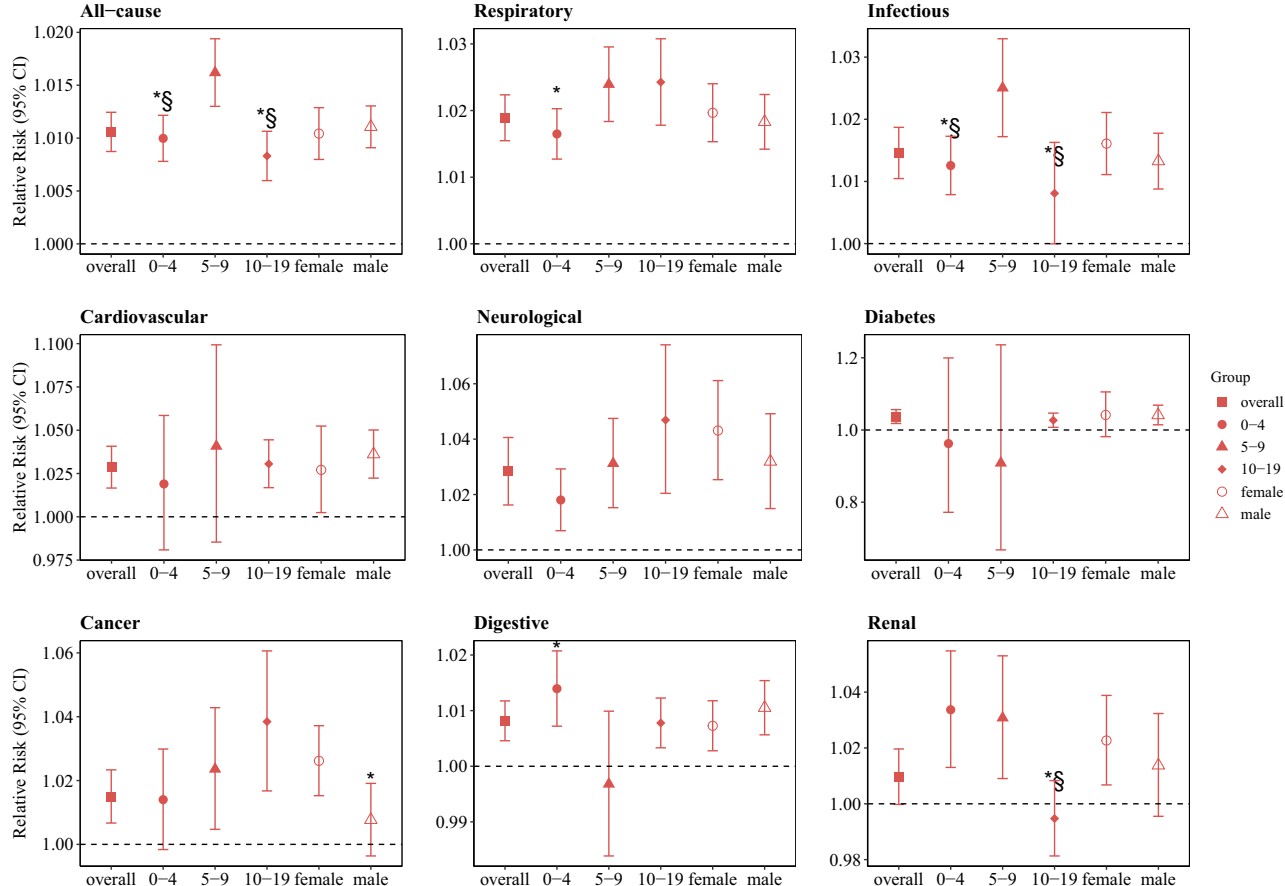

**Fig. 4 | The cumulative relative risks of all-cause and cause-specific hospital admissions in children and adolescents aged 0–19 years associated with 10 μg/m³ increase in LFS PM$_{2.5}$ during lag 0–1 day in 1012 communities.** Point shapes indicate subgroups: solid square = overall; solid circle = 0–4 years; solid triangle = 5–9 years; open triangle = 10–19 years; open circle = female; open triangle = male. Data are presented as estimated relative risks (RRs) with 95% confidence intervals (CIs). Central points represent estimated RRs; error bars represent corresponding 95% CIs. Estimates were derived from a two-stage time-series analysis. *$P\_unadjusted$ < 0.05, P values are not adjusted for multiple comparisons. §$P\_adjusted$ < 0.05, P values are adjusted for multiple comparisons using the FDR method. The adjustment was implemented using the Benjamini–Hochberg (BH) method via the p.adjust function (method = "fdr") in R. All statistical tests were two-sided.

Significant effect modification by age was observed for all-cause and infectious admissions in our study. Compared with children aged 5–9 years, those aged 0–4 years and 10–19 years showed lower hospital admission risks. Higher outdoor activity levels in children aged 5–9 years likely increased their exposure to LFS PM$_{2.5}$ compared to younger children, while their relatively immature respiratory and immune systems may increase their susceptibility to its health effects compared to older children[38]. However, these age differences should be interpreted with caution, as some age-specific associations were not statistically significant, possibly due to limited sample sizes for certain diseases.

Our study has several strengths. The main strength is that, to our knowledge, it was the most comprehensive multi-country/territory study concentrating on an extensive range of child morbidities associated with LFS PM$_{2.5}$. Compared with previous evidence concentrating on a specific region or a specific morbidity, we provided a more comprehensive exploration covering a large sample size and extensive morbidity outcomes. Another strength was the stratified analyses by age, sex, and area-level economic indices, which allowed identification of subgroups of children more vulnerable to LFS PM$_{2.5}$.

However, several limitations should be noted. Although our study covered a large sample size from seven countries/territories, its global representativeness remains limited, particularly for low-income countries. The focus on hospital admissions, rather than emergency department visits, may lead to different risk estimates

and restrict the scope of our findings. Additionally, the estimation of LFS PM$_{2.5}$ from GEOS-Chem introduces greater uncertainty compared to total PM$_{2.5}$, as LFS PM$_{2.5}$ relies on model simulations and assumptions about emission sources and chemical transport[1]. Fire emission inputs were based on the Global Fire Emissions Database (GFED4.1s), a widely used and comprehensive dataset. However, it may under-detect small fires, such as agricultural biomass burning, due to limitations in satellite resolution and interference from cloud cover or vegetation. This may lead to some degree of exposure misclassification. Because exposure estimates were aggregated to the community level using population-weighted averages, such misclassification is likely to be Berkson error, which tends to bias effect estimates toward the null. Therefore, the reported associations may underestimate the true health effects of LFS PM$_{2.5}$. Moreover, the use of community-level, rather than individual LFS PM$_{2.5}$ exposure, could also result in exposure misclassification, potentially attenuating effects to null. Future studies should cover a broader geographic range, consider individual-level data, and further increase the exposure estimate accuracy.

In summary, LFS PM$_{2.5}$ is associated with increased risks of all-cause and cause-specific child morbidities with heterogeneity across geography, age, and area-level economic factors. Regarding landscape fires, targeted measures and further investigations for children are expected to supplement knowledge and reduce morbidity risks, especially for more vulnerable subgroups.

## Methods

The study was approved by the Monash University Human Research Ethics Committee (Project ID: 24439). The data were collected by local health authorities in accordance with relevant ethical and legal regulations in each country.

### Hospital admission data

Daily counts of hospital admissions were obtained from New South Wales (Australia), Brazil, Canada, Chile, New Zealand, Taiwan, and Thailand during 2000 to 2019. These hospital admission data covered nearly the entire population in most regions, except for Brazil and Chile, where coverage was approximately 70–80% (Table S1). Details of the data collection and organization procedure for each community are provided in Method S1 in supplementary files. In summary, we aggregated and integrated a multi-country dataset of daily age-, sex-, and cause-specific hospital admissions for infectious diseases, cancer, diabetes, neurological system disorders, cardiovascular diseases, respiratory diseases, digestive diseases, and renal diseases from 1012 communities across the seven countries/territories, for subsequent analysis. These disease categories were selected due to their biological plausibility and previous associations with LFS $PM_{2.5}$ exposure in adult populations. LFS $PM_{2.5}$ exposure may lead to systemic inflammation, oxidative stress, immune dysfunction, and neurotoxicity, potentially increasing risks for multiple organ systems[7,20–24].

We used community-level hospital admission data because they provided finer spatial resolution than city- or country-level data. The study regions varied in spatial community sizes, with a median area of 5384.1 km$^2$ and a median population of 126,708 (Table S1). This approach helps capture within-country variations in susceptibility to LFS $PM_{2.5}$, such as categorizing each country based on the SES level of communities, which allows the identification of vulnerable and high-risk communities. Understanding these differences is crucial for targeted interventions and prevention strategies at local levels.

### LFS $PM_{2.5}$ estimation

We obtained daily LFS $PM_{2.5}$ concentrations from our previous study, which estimated fire-sourced air pollution at a 0.25° × 0.25° (≈28 × 28 km) spatial resolution using a three-step approach[1].

First, fire-sourced and total $PM_{2.5}$ concentrations were estimated using GEOS-Chem by running two simulations: one with fire emissions and one without fire emissions. The difference between these simulations provided the fire-sourced $PM_{2.5}$ concentration at 2.5° × 2.5° resolution. Fire emission data were obtained from the Global Fire Emissions Database (GFED4.1s)[39], which captured emissions from six fire sources: boreal forest fires, tropical forest fires, savanna/grassland/shrubland fires, temperate forest fires, peatland fires, and agricultural waste burning. These emissions were derived from satellite retrievals of burned areas and active fire detections. Then fire-sourced and total $PM_{2.5}$ estimates were downscaled from 2.5° × 2.5° to 0.25° × 0.25° using inverse distance weighted (IDW) spatial interpolation[40].

Second, the downscaled estimates were calibrated using a random forest machine learning model, incorporating ground-based observations from 5661 monitoring stations across 73 countries. This step reduced bias and improved accuracy in $PM_{2.5}$ estimates.

Third, the fire-sourced fraction of $PM_{2.5}$ from GEOS-Chem was then applied to the bias-corrected total $PM_{2.5}$ estimates to obtain the final fire-sourced $PM_{2.5}$ dataset at a 0.25° × 0.25° resolution. These estimates were validated using spatial tenfold cross-validation, which demonstrated high accuracy for daily total $PM_{2.5}$ estimates, with an R$^2$ of 0.91 and a root mean squared error (RMSE) of 8.47 μg/m$^3$.

We selected a 0.25° × 0.25° spatial resolution for LFS $PM_{2.5}$ exposure assessment, which aligns well with the spatial scale of most communities in our study. Rather than using single-point estimates (e.g., from the community centroid), we calculated population-weighted averages of LFS $PM_{2.5}$ across all grid cells within each

community. The calculation method is described in the following section titled "Estimating community-level population-weighted values for exposures." This approach enhances the representativeness of exposure estimates by accounting for both spatial variability and population distribution.

### Meteorological data

Daily temperature and 2 m dewpoint temperature data at a spatial resolution of 0.25° × 0.25° were derived from the European Centre for Medium-Range Weather Forecasts Reanalysis v5 (ERA5)[41]. Since ERA5 provides 2 m dewpoint temperature, which is a measure of air humidity, we calculated daily relative humidity using temperature and dewpoint temperature to ensure consistency across study regions[42].

### Estimating community-level population-weighted values for exposures

Daily community-level population-weighted values for LFS $PM_{2.5}$, temperature, and relative humidity were calculated by averaging the values from all grid cells within each community, weighted by the population counts of each grid cell. These values were further matched with hospital admission counts by date and community.

### Demographic and socioeconomic data

Population data at about 1 km grid-cell resolution were obtained from the Gridded Population of the World (GPW, version 4)[43]. Gridded gross domestic product (GDP)[44] was then obtained to calculate local GDP per capita by dividing local GDP by the local population at grid scale. GDP per capita was then categorized by tertiles as low, middle, or high groups.

Country income classification was obtained from the World Bank (World Bank Country and Lending Groups – World Bank Data Help Desk). Of the countries involved in this study, Australia, Canada, New Zealand, and Taiwan were high-income countries/territories, while Brazil, Chile, and Thailand were upper-middle-income countries.

### Statistical analysis

To examine the associations between LFS $PM_{2.5}$ and all-cause and cause-specific hospital admission, a two-stage time-series analytical approach was employed. In the first stage, a quasi-Poisson regression distributed lag non-linear model was used for each community to obtain the community-specific effect estimates for LFS $PM_{2.5}$-hospital admission association. The model incorporated a cross-basis function for LFS $PM_{2.5}$ (0–14 lag days) to account for both non-linear exposure–response relationships and delayed effects. We adjusted potential confounders, including temperature (0–21 lag days) and relative humidity (0–7 lag days). Additionally, natural cubic splines for year and an indicator for the day of the week (DOW) were included to control for long-term and seasonal trends. Details of the model are shown in Method S2 in supplementary file. In the second stage, the community-specific effects from the first stage were pooled to estimate the associations between LFS $PM_{2.5}$ and hospital admission risk, using random effects meta-analysis[45]. The non-linear exposure–response relationship between LFS $PM_{2.5}$ and hospital admission risk was modeled using a natural cubic spline with two knots at the 25th and 75th of the mean LFS $PM_{2.5}$ distribution across all the communities in the study[46,47]. We used the false discovery rate (FDR) adjustment to control for Type I error due to multiple testing. The adjustment was implemented using the Benjamini–Hochberg (BH) method via the p.adjust function (method = "fdr") in R.

To identify the potential modification of the associations between LFS $PM_{2.5}$ and hospital admission risk, stratified analyses and Wald-type test[48] were employed by sex (female or male), age group (0–4, 5–9, or 10–19 years), and socioeconomic status (local GDP per capita and country income class), shown in Method S3 in supplementary file.

To ensure the robustness of the results, the following sensitivity analyses were performed: (1) placing the knots for natural cubic spline of LFS PM$_{2.5}$ at 10th and 90th or 20th and 80th centiles of the mean exposure distribution; (2) changing the df for meteorological variables: temperature from 4 to 3 and 5, relative humidity from 3 to 4 and 5; (3) changing the df for time trends from 7 to 8 and 9. All analyses were performed in R software (version 4.1.1), with dlnm[49],mixmeta[48], splines, lubridate, and ggplot2 packages. A two-sided p-value < 0.05 was considered statistically significant.

## Ethical declarations
This study was approved by the Monash University Human Research Ethics Committee (ID 24439).

## Reporting summary
Further information on research design is available in the Nature Portfolio Reporting Summary linked to this article.

## Data availability
The health and hospital admission data used in this study are derived from multiple countries. Due to ethical and legal restrictions, these data cannot be made publicly available. The data are available under restricted access for reasons of participant confidentiality and national data-sharing agreements. Access can be requested by contacting the corresponding authors, Y.G. and S.L. Requests must include a brief research proposal and documentation of relevant ethical approvals. All requests will be reviewed in a timely manner, and, if approved, data access will be granted for non-commercial research purposes for a fixed period as specified in a data-sharing agreement. Source data underlying the figures are provided at GitHub: https://github.com/Shuang0601/Landscape-fire-and-children-morbidity and archived on Zenodo: https://doi.org/10.5281/zenodo.16238692.

## Code availability
The code used in this study is available at GitHub: https://github.com/Shuang0601/Landscape-fire-and-children-morbidity and archived on Zenodo: https://doi.org/10.5281/zenodo.16238692.

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

## Acknowledgements

S.Z. was supported by Monash Faculty of Medicine Nursing and Health Science (FMNHS) Early Career Postdoctoral Fellowships 2025. R.X. was supported by the VicHealth Postdoctoral Research Fellowships 2022. P.Y. was supported by Monash Faculty of Medicine Nursing and Health Science (FMNHS) Early Career Postdoctoral Fellowships 2023. G.C. was supported by a Monash Early Career Postdoctoral Fellowship and an Australian National Health and Medical Research Council (NHMRC) Centre for Safe Air Postdoctoral Research Fellowship. T.Y. was supported by China Scholarship Council funds (number 201906320051). B.W. was supported by China Scholarship Council funds (number 202006010043). Y.W. was supported by China Scholarship Council funds (number 202006010044). W.H. was supported by China Scholarship Council funds (number 202006380055). Z.Y. was supported by the Monash Graduate Scholarship and the Monash International Tuition Scholarship. W.Y. was supported by the Monash Graduate Scholarship and Monash International Tuition Scholarship. K.J. is supported by China Scholarship Council funds (number 202006240087). Z.X. is supported by the Monash Graduate Scholarship and the Monash International Tuition Scholarship. K.T. is supported by the National Research Council of Thailand (N33A650979). Y.G. was supported by a Leader Fellowship (APP2008813) of the NHMRC. S.L. was supported by an Emerging Leader Fellowship of the NHMRC (APP2009866).

## Author contributions

Y.G. and S.L. contributed equally to the correspondence work. Y.G. and S.L. conceptualized the study. S.Z., Y.G., and S.L. designed the methodology. S.Z. performed the analysis, produced the original figures, and drafted the manuscript. Yiwen Z., Z.Y., R.X., Z.X., and Y.G. contributed to the analysis. R.X. coordinated the multi-country data collection and cleaning. R.X., M.A., L.M., F.H.J., S.H., M.S.C., Y.L.G., J.H., W.K., L.K., E.L., G.M., P.M., G.M., P.H.S., and K.T. contributed to data collection. Yiwen Z., Z.Y., R.X., W.H., Y.W., Y.L., W.Y., P.Y., K.J., T.Y., B.W., and Yuxi Z. performed the multi-country data cleaning. All authors contributed to the critical revision of the manuscript. S.L. and Y.G. provided administrative, technical, and material support, supervision, and mentorship. Each author contributed important intellectual content during manuscript drafting or revision and accepts accountability for the overall work by ensuring that questions pertaining to the accuracy or integrity of any portion of the work are appropriately investigated and resolved. All authors approved the final version of the manuscript. S.L. and Y.G. are the study guarantors. The corresponding authors attest that all listed authors meet authorship criteria and that no others meeting the criteria have been omitted.

## Competing interests

The authors declare no competing interests.

## Additional information

[1]Climate Air Quality Research Unit, School of Public Health and Preventive Medicine, Monash University, Melbourne, VIC, Australia. [2]Chongqing Emergency Medical Center, Chongqing University Central Hospital, School of Medicine, Chongqing University, Chongqing, China. [3]School of Life and Environmental Sciences, University of Sydney, Sydney, NSW, Australia. [4]School of Earth and Atmospheric Sciences, Queensland University of Technology, Brisbane, QLD, Australia. [5]Menzies Institute for Medical Research, University of Tasmania, Hobart, TAS, Australia. [6]Department of Public Health, University of Otago, Wellington, New Zealand. [7]Department of Pathology, School of Medicine, University of São Paulo, São Paulo, Brazil. [8]Environmental and Occupational Medicine, National Taiwan University College of Medicine and National Taiwan University Hospital, Taipei, Taiwan, ROC. [9]Faculty of Health and Medical Sciences, University of Western Australia, Perth, WA, Australia. [10]Social and Environmental Medicine, Faculty of Tropical Medicine, Mahidol University, Nakhon Pathom, Thailand. [11]School of Public Health, The University of Sydney, Sydney, NSW, Australia. [12]School of Epidemiology and Public Health, University of Ottawa, Ottawa, ON, Canada. [13]School of Clinical Medicine, University of New South Wales, Sydney, NSW, Australia. [14]School of Medicine, University of the Andes (Chile), Santiago, Chile. ✉e-mail: yuming.guo@monash.edu; shanshan.li@monash.edu

