## [Peer Review file · Nature Communications]

Impact of global short-term landscape fire sourced PM2.5 exposure on child cause-specific morbidity: a study in multiple countries and territories

Corresponding Author: Professor Shanshan Li

Version 0:

Reviewer comments:

Reviewer #1

(Remarks to the Author)

This study by Zhou et al. evaluates the associations between acute LFS PM2.5 exposure and child hospital admissions. This is a highly significant topic, and the authors have conducted extensive analyses and validations. However, the study presents results across multiple dimensions, and the interpretation of these findings, particularly from mechanistic and theoretical perspectives, requires further strengthening.

(1) The hospital admission data were collected from seven regions: Australia, Brazil, Canada, Chile, New Zealand, Taiwan, and Thailand. How representative are these regions of fire-prone areas? Do they encompass global regions with high fire-related PM2.5 exposure? This directly affects the global applicability of the study's findings.

(2) The associations between exposure and eight cause-specific hospital admissions were evaluated across 1,012 communities, raising concerns about the issue of multiple comparisons. With so many comparisons in a single study, it is statistically unlikely that no significant associations would be identified under a 95% confidence interval. The authors should address this concern by controlling the Type I error, rather than merely acknowledging it as a limitation.

(3) The associations between PM2.5 exposure and some target diseases, such as cancer and type I diabetes, lack sufficient biological or medical support. The authors did not provide evidence from previous studies on potential mechanisms in the Introduction. Additionally, limited discussion was provided in the Discussion section to explain the varying shapes of the exposure-response function. The authors are encouraged to offer more substantial evidence on the plausible mechanisms linking acute exposure to the target diseases.

(4) How should the differences in RR across age groups be understood? For instance, in the case of cardiovascular diseases, the RR for the overall group indicates that children aged 5–9 years have a higher RR than those aged 0–4 years, while the RR for the 10–19 age group is lower than that for the 0–4 age group. The authors should provide a mechanistic explanation for these age-specific variations.

(5) Similar issues are for income levels. These results are somewhat more intuitive, with lower-income regions showing higher RRs compared to higher-income regions. The authors are advised to supplement the discussion with relevant insights to further explain these findings.

(Remarks on code availability)

Reviewer #2

(Remarks to the Author)

Landscape fires are increasing over the years under the global warming and understandign the health risks of landscape fire-PM2.5 is critical to develop appropriate adaptaiton and mitigaiton strategies. This work examines the association between short-term exposdure to LFS-PM2.5 and cause-specific hostpial admissions (for children) in seven countries using a time-series design. While numerous time-series studies exist in the literature, analysis with source-specific Pm2.5 is generally lacking. Overall, the study is quite interesting and presents novel results. However, it lacks clarity in the narrative and sufficeint details to fully interpret the results. After addressing the comments (provided below), the manuscript mau be considered.

Summary

Findings should have more quantitative estimates; currently it just says - significant association. What are the risks? How does it vary across diseases / which diseases are more affected by LFS-PM2.5? Which subgroups are more vulnerable?

Methods - lack sufficient details in the manuscript.

Are the validation statistics for daily Pm2.5 estimates? if not, this should be reported for daily scale as the time-series used daily exposure.

Are you using the data reported in the reference no 1? If yes, provide more details about the exposure across your study populations. There is only a few lines about exposure in 3 countries - whether these are for the communities from where health data are obtained is not clear. What about other regions?

The method to estimate LFS-PM2.5 is not clear. Are you using the mass fractions of LFS-Pm2.5 from geos-chem and integrating with 0.25 degree downscaled adjusted Pm2.5?

How large are the communities? Since the admission data are from specific hospitals, how large is their catchment area? cities? beyond? Is 0.25 degree spatial scale enough?

The statistical analysis section in the main manuscript should briefly discuss the model with details in Method S2. You have adjusted for temperature. Is LFS-PM2.5 concentration correlated with temperature? If yes, there could be over-adjustment in the model.

Results

Results are reported for 0 and 1 lag. There are almost negligible effects for lag greater 2 or more days. On the contrary, many time series studies showed effect showing lag for several days. Is it because the LFS generated Pm2.5 gets diluted to have any significant effect?

For some outcome like digestive, the effect actually increases from lag of 4 days and above - why?

Fig 3 mentions wildfire-PM2.5. Are you considering only wildfire or all types of fire?

Xu et al (2023) where LFS-PM2.5 data were generated showed larger uncertainty than for PM2.5. It's not directly possible to evaluate sector-specific PM2.5, but the fact that the effects could have larger errors than reported due to the error in exposure should be acknowledged.

For pooled estimates, there is contrast in the observed association - for some diseases some age groups do not show significant association. Any plausible explanation?

(Remarks on code availability)

The code will be available from the authors. I did not ask for the codes and assumed that the code works alright.

Reviewer #3

(Remarks to the Author)

The manuscript titled "Impact of global short-term landscape fire sourced PM2.5 exposure on child cause-specific morbidity: a study in multiple countries and territories" presented a multi-country, time-series study investigating the short-term health impacts of landscape fire-sourced (LFS) PM2.5 on children and adolescents. It analyzed cause-specific hospital admission data from 1012 communities in seven countries/territories (Australia, Brazil, Canada, Chile, New Zealand, Taiwan, and Thailand) from 2000 to 2019. The study stated that short-term exposure to LFS PM2.5 increased hospital admissions for various diseases in children and adolescents. The manuscript is well-structured with robust evidence. I suggest to publish it after moderate revision.

Major comments:

1. "Associations between LFS PM2.5 and hospital admissions were first evaluated at the community level..." Please address the importance to evaluate at community level, what makes it different or more advanced/reasonable compared with other levels (e.g. cities, countries, towns)
2. In the introduction section, you did not summarize any other studies (regional- or global-wise) on short-term impacts of LFS PM2.5. Does impacts of LFS PM2.5 mentioned in introduction referred to long-term impacts? I suggest to compare results of other studies on short-term impacts of LFS PM2.5 in introduction or discussion.
3. I suggest to summarize all the diseases analyzed in results in methods section, and provide reasons why they are selected in the study. Are their associations with LFS PM2.5 mentioned in previous studies?
4. "The highest risks occurred on the current day (lag0) of LFS PM2.5 exposure and decreased afterwards, with significantly positive associations on lag0 and lag1 days." Is the result consistent with previous studies on lag effect as well?
5. In discussion section: "We observed that LFS PM2.5 was associated with increased risks of all-cause, respiratory, infectious, cardiovascular, neurological, diabetes, cancer, and digestive hospital admissions among children." Do you mean short-term exposure to LFS PM2.5 here? If so, how does it increase chronic diseases as cancer, diabetes?
6. In Line 298: "inverse distance weighted spatial interpolation", I suggest to provide a reference? The fire emission inventory used in GEOS-CHEM model should also be stated here since it is highly related with the study.
7. In Line 309: "Daily relative humidity was calculated from daily temperature and dewpoint temperature." I believe ERA5

provided relative humidity data as well. Is there a specific reason you need to calculated by yourself?

(Remarks on code availability)

Version 1:

Reviewer comments:

Reviewer #1

(Remarks to the Author)

The authors have addressed all my comments so I recommend the manuscript to be accepted for publication.

(Remarks on code availability)

Reviewer #3

(Remarks to the Author)

After careful evaluation of the revised manuscript entitled " Impact of global short-term landscape fire sourced PM2.5 exposure on child cause-specific morbidity: a study in multiple countries and territories", I find the authors have adequately addressed all review comments. The manuscript now meets the journal's publication standards and is recommended for acceptance. I have just one question:

- 1.The study explored the short-term landscape fire sourced PM2.5 exposure on child cause-specific morbidity. Why the chronic disease (i.e., cancer and diabetes) are selected in this study?
2. fire sourced PM2.5 exposure was estimated using GFED4.1s, some small fire (i.e., agricultural biomass burning) might not be detected, which might underestimate the result. Please discuss this issue.
- 3.L439: "changing the df for meteorological variables: temperature from 4 to 3 and 5, RH from 3 to 4 and 5. 3)". Should the later reference to 5.3 be corrected to df=5?"

(Remarks on code availability)

Reviewer #4

(Remarks to the Author)

1. Why these countries? Which areas are at higher risk of LFS PM2.5? Lower risk? This could be included in the introduction.
2. Can you explain the cardiovascular hospital admissions in children? This is not necessarily expected in this age group and warrants further explanation.
3. The last sentence of the first paragraph of the methods needs references (lines 346-348).
4. For the description of the LFS PM2.5 estimation, please provide a referece to the previous study (Lines 359-361).
5. A recommendation is to also include the median population size of the communities here as well. (Line 352)
6. How was the false discovery rate adjustment conducted?
7. Is there a typo for the RH in line 441?

(Remarks on code availability)

REVIEWER COMMENTS

Reviewer #1 (Remarks to the Author):

This study by Zhou et al. evaluates the associations between acute LFS PM_{2.5} exposure and child hospital admissions. This is a highly significant topic, and the authors have conducted extensive analyses and validations. However, the study presents results across multiple dimensions, and the interpretation of these findings, particularly from mechanistic and theoretical perspectives, requires further strengthening.

(1) The hospital admission data were collected from seven regions: Australia, Brazil, Canada, Chile, New Zealand, Taiwan, and Thailand. How representative are these regions of fire-prone areas? Do they encompass global regions with high fire-related PM_{2.5} exposure? This directly affects the global applicability of the study's findings.

Response:

Thanks for your comments. Landscape fire smoke can travel hundreds or even thousands of kilometres, impacting air quality and health far from the fire source ^{1,2}. During long-range transport, LFS PM_{2.5} undergoes chemical transformation, which can increase its oxidative and proinflammatory components ³, potentially exacerbating health impacts even in regions without direct fire activity. Therefore, assessing health risks in both fire-prone and non-fire-prone regions is essential for fully understanding the impacts of LFS PM_{2.5} exposure.

Our study includes both fire-prone (Brazil, Chile, Australia, Canada, and Thailand) and non-fire-prone (New Zealand and Taiwan) regions. While wildfires and agricultural burning frequently contribute to high fire-related PM_{2.5} exposure in fire-prone regions, non-fire-prone areas can still experience substantial transboundary smoke transport. By incorporating both types of regions, we capture the health effects of both direct and long-range smoke exposure. We have also added explanations for geographical heterogeneity in the Discussion. However, our study does not include data from low-income countries, which limits its global representativeness. Future research incorporating these regions will further improve our understanding of these health effects globally. We have acknowledged this limitation in the Discussion. Please see Lines 222-233, 267-269.

“We observed geographical heterogeneity in hospital admissions linked to LFS PM_{2.5} across different countries/territories, including both fire-prone and non-fire-prone regions. These differences may stem from variations in vegetation type ³¹, LFS events pattern ³², population demographics, environmental policies, healthcare infrastructure, and help-seeking behaviors ³³. Interestingly, those non-fire-prone regions like New Zealand and Taiwan showed relatively higher risks for all-cause and respiratory hospital admission associated with LFS PM_{2.5} than those fire-prone regions. This may be due to the chemical transformation of LFS during long-range transport, which can increase oxidative and proinflammatory components of LFS PM_{2.5} ³⁴. Additionally, these regions experience LFS events less frequently, thus they may lack adequate response systems, leading to higher hospital admissions when exposure occurs ³⁵. These findings highlight the need for region-specific public health interventions tailored to local environmental conditions and vulnerabilities.”

“However, several limitations should be noted. Although our study covered a large sample size from seven countries/territories, its global representativeness remains limited, particularly for low-income countries.”

(2) The associations between exposure and eight cause-specific hospital admissions were evaluated across 1,012 communities, raising concerns about the issue of multiple comparisons. With so many comparisons in a single study, it is statistically unlikely that no significant associations would be identified under a 95% confidence interval. The authors should address this concern by controlling the Type I error, rather than merely acknowledging it as a limitation.

Response:

We acknowledge that conducting multiple comparisons across eight cause-specific hospital admissions and various subgroups increases the likelihood of Type I error. To address this issue, we applied the False Discovery Rate (FDR) adjustment to control for multiple testing. After adjustment, most associations remained statistically significant, while the association with renal hospital admissions and the effect modification by sex became nonsignificant. This adjustment enhances the robustness of our findings and reduces the risk of false-positive results. We have revised the Methods, Results, and Discussion accordingly. Please see Methods in Lines 106, 111-112, 114-115, 129, 136-137, 384-385, Table 1, Figure 4, and Figure S1-S2.

“After adjusting for multiple comparisons using false discovery rate (FDR) method,…”

“The association with renal disease admissions [1.010 (1.000, 1.020)] was no longer statistically significant after FDR adjustment. (Table 1).”

“Geographical heterogeneity existed in the conditions sensitive to LFS PM_{2.5} across different countries/territories after FDR adjustment (Table 1).”

“Stratified analyses by age and sex after FDR adjustment are shown in Figure 4 and Table 1.”

“Similar significant effect modification by local GDP per capita and country income class was observed after FDR adjustment (Table 1, Figure S1, S2).”

“.... We used the false discovery rate (FDR) adjustment to control for Type I error due to multiple testing.”

(3) The associations between PM_{2.5} exposure and some target diseases, such as cancer and type I diabetes, lack sufficient biological or medical support. The authors did not provide evidence from previous studies on potential mechanisms in the Introduction. Additionally, limited discussion was provided in the Discussion section to explain the varying shapes of the exposure-response function. The authors are encouraged to offer more substantial evidence on the plausible mechanisms linking acute exposure to the target diseases.

Response:

In the Introduction, we have incorporated evidence on how LFS PM_{2.5} induces systemic inflammation, oxidative stress, and immune dysfunction, which are linked to multiple organ systems. This supports the plausibility of associations beyond respiratory diseases.

In the Discussion, we have expanded the mechanistic explanations. We also explain the mechanisms for cancer and type I diabetes, highlighting that short-term exposure to LFS PM_{2.5} may exacerbate complications in paediatric patients with pre-existing diabetes or cancer rather than triggering new-onset cases. Additionally, we have provided further discussion on the exposure-response function and potential reasons for variations in risk at different LFS PM_{2.5} levels. Please see Lines 39-47 in the Introduction and Lines 179-220 in the Discussion.

“Inhaled LFS PM_{2.5} can deposit deep into the lungs to trigger respiratory inflammation and oxidative stress, leading to respiratory damage and increased susceptibility to infections⁶. Its toxic components, such as ultrafine particles and heavy metals, can enter the bloodstream to cause systemic inflammation, oxidative stress, and immune dysfunction⁷. These biological pathways suggest that LFS PM_{2.5} may increase risks for multiple organ systems, as widely documented in adults^{4,5}.

Children exhibit greater vulnerability to LFS PM_{2.5} possibly due to behaviours such as spending more time outdoors and the immaturity of their respiratory and immune systems⁸.”

“The observed associations between LFS PM_{2.5} exposure and increased hospital admissions can be explained by its biological mechanisms. Upon inhalation, LFS PM_{2.5} deposits deep into the respiratory tract and triggers inflammation and oxidative stress, which can lead to lung damage and increased susceptibility to infections²⁰⁻²². Some toxic components (e.g., ultrafine particles and heavy metals) can enter the bloodstream and induce systemic inflammation that may lead to cardiovascular²³, metabolic, and other diseases²⁴. Beyond the lungs, LFS PM_{2.5} can reach the brain via the olfactory nerve or bypassing the blood-brain barrier, which will lead to neuroinflammation and increase the risks of neurological disorders²⁵. LFS PM_{2.5} can also be ingested through contaminated food and water or transported to the gut via mucociliary clearance. Once in the digestive system, it may disrupt gut microbiota, increase intestinal permeability, and promote inflammation, resulting in digestive diseases²⁶. Additionally, fire smoke exposure weakens immune function, increasing susceptibility to respiratory and systemic infections²⁷. These mechanisms provide a biological explanation for the widespread health impacts observed in our study.”

“The lag patterns observed in our study align with these mechanistic pathways. The observed largest effects at lag 0-1 days suggest an acute inflammation and oxidative stress response to LFS PM_{2.5} exposure, consistent with previous studies¹¹⁻¹⁴. The attenuation of these associations at longer lags may result from the dilution and dispersion of LFS PM_{2.5}, leading to lower exposure levels and reduced direct impact on health³. However, digestive diseases showed a secondary rise in risk around lag 7, potentially reflecting a combination of immediate inflammatory responses and delayed gastrointestinal effects through mucociliary clearance and ingestion pathways²⁶. This prolonged effect highlights the need for clinicians to monitor for delayed-onset symptoms and for healthcare systems to allocate resources accordingly to ensure adequate medical support beyond the immediate exposure period.

The exposure-response relationships observed further supported these mechanistic pathways. Risks for all-cause, respiratory, and infectious hospital admissions increased steadily from low levels of LFS PM_{2.5} exposure, indicating that even minimal exposure exacerbated respiratory inflammation and immune responses⁴. Neurological diseases

showed a more gradual increase in risk at very low exposure levels, which may reflect both the ability of LFS PM_{2.5} to reach the brain even at low exposure levels and the slower progression of neuroinflammation compared to acute respiratory and immune responses^{28,29}. In contrast, cardiovascular (mainly peripheral vascular diseases in children of this study), diabetes (mainly Type I diabetes), cancer (mainly leukemia), digestive (mainly acute pancreatitis), and renal (mainly renal tubulointerstitial diseases) risks increased only at higher concentrations, around 15-20 µg/m³, suggesting more sustained or higher exposure is needed to trigger systemic effects through oxidative stress and inflammation. Especially for pediatric patients with pre-existing diabetes or cancer, short-term LFS PM_{2.5} may compromise their immune function and exacerbate their pollution-induced complications rather than triggering new-onset cases^{24,30}. Further research is needed to understand these exposure-response relationships, particularly at higher LFS PM_{2.5} concentrations.”

(4) How should the differences in RR across age groups be understood? For instance, in the case of cardiovascular diseases, the RR for the overall group indicates that children aged 5–9 years have a higher RR than those aged 0–4 years, while the RR for the 10–19 age group is lower than that for the 0–4 age group. The authors should provide a mechanistic explanation for these age-specific variations.

Response:

After applying the False Discovery Rate (FDR) adjustment, the age-specific variation for cardiovascular diseases was no longer statistically significant. However, we still observed significant effect modification by age for all-cause and infectious disease hospital admissions. Compared with children aged 5-9 years, those aged 0-4 years and 10-19 years showed lower hospital admission risks.

We have expanded our discussion on age-related susceptibility. Higher outdoor activity levels in children aged 5-9 years likely increased their exposure to LFS PM_{2.5} compared to younger children, while their relatively immature respiratory and immune systems may increase their susceptibility to its health effects compared to older children. Please see Lines 250-257.

“Significant effect modification by age was observed for all-cause and infectious admissions in our study. Compared with children aged 5-9 years, those aged 0-4 years and 10-19 years showed lower hospital admission risks. Higher outdoor activity levels in children aged 5-9 years likely increased their exposure to LFS PM_{2.5} compared to younger children, while their relatively immature respiratory and immune systems may increase their susceptibility to its health effects compared to older children³⁸. However, these age differences should be interpreted with caution, as some age-specific associations were not statistically significant, possibly due to limited sample sizes for certain diseases.”

(5) Similar issues are for income levels. These results are somewhat more intuitive, with lower-income regions showing higher RRs compared to higher-income regions. The authors are advised to supplement the discussion with relevant insights to further explain these findings.

Response:

We have added explanations for the SES modification effect, including higher LFS PM_{2.5} exposure levels, limited access to protective measures, poorer baseline health, inadequate healthcare access, and weaker policy implementation in lower-SES regions, which collectively lead to their higher risks of hospital admissions. Please see Lines 235-248.

“Socioeconomic status (SES) also influences the health impacts of LFS PM_{2.5}, with lower-SES regions showing higher risks of all-cause and digestive-related hospital admissions. Several factors may explain this disparity. First, lower-SES regions often experience higher LFS PM_{2.5} concentrations while people in these areas may have fewer resources to adopt protective measures such as indoor air filtration systems¹. Second, children in lower-SES areas tend to have poorer baseline health status and limited healthcare programs (e.g., routine health check-ups, vaccinations, and comprehensive public health education programs)³⁶, making them more vulnerable to LFS PM_{2.5}. Third, inadequate healthcare access can lead to delayed treatment, worsening disease severity. Fourth, limited policy implementation addressing wildfire events and air pollution in lower-SES areas further exacerbates health risks³⁷. This finding underscores the need for targeted interventions such as improving indoor air quality through air filtration systems, increasing public awareness through education programs, and strengthening policies to mitigate the disproportionate burden of LFS PM_{2.5} in lower SES regions.”

Reference:

1. Zhou, S., Xu, R., Chen, G., Yu, P. & Guo, Y. Where is the boundary of wildfire smoke? The Innovation Medicine 1, 100024, doi:10.59717/j.xinn-med.2023.100024 (2023).
2. Xu, R. et al. Wildfires, Global Climate Change, and Human Health. The New England journal of medicine 383, 2173-2181, doi:10.1056/NEJMSr2028985 (2020).
3. Wong, J. P. S. et al. Effects of Atmospheric Processing on the Oxidative Potential of Biomass Burning Organic Aerosols. Environmental science & technology 53, 6747-6756, doi:10.1021/acs.est.9b01034 (2019).

Reviewer #2 (Remarks to the Author):

Landscape fires are increasing over the years under the global warming and understanding the health risks of landscape fire-PM_{2.5} is critical to develop appropriate adaptation and mitigation strategies. This work examines the association between short-term exposure to LFS-PM_{2.5} and cause-specific hospital admissions (for children) in seven countries using a time-series design. While numerous time-series studies exist in the literature, analysis with source-specific Pm_{2.5} is generally lacking. Overall, the study is quite interesting and presents novel results. However, it lacks clarity in the narrative and sufficient details to fully interpret the results. After addressing the comments (provided below), the manuscript may be considered.

Summary

1. Findings should have more quantitative estimates; currently it just says - significant association. What are the risks? How does it vary across diseases / which diseases are more affected by LFS-PM_{2.5}? Which subgroups are more vulnerable?

Response:

Thanks for your comments. We have revised the Findings section in the Abstract to include quantitative estimates, reporting the relative risks with 95% confidence intervals across different diseases and highlighting the variations in exposure-response patterns. We have also specified the most vulnerable subgroups. Additionally, we have updated the Results section with more detailed quantitative estimates to provide a clearer picture of the associations. Please see Lines 15-23 in the Abstract and Lines 106-112 in the Results section.

“Each 10 µg/m³ increase in LFS PM_{2.5} was associated with increased risks of all-cause [Relative risk: 1.011 (95% confidence interval: 1.009, 1.012)] , respiratory [1.019 (1.015, 1.022)], infectious [1.015 (1.010, 1.019)], cardiovascular [1.029 (1.017, 1.041)], neurological [1.028 (1.016, 1.041)], diabetes [1.037 (1.018, 1.057)], cancer [1.015 (1.007, 1.023)], and digestive [1.008 (1.005, 1.012)] hospital admissions. Risks for respiratory, infectious, and neurological admissions increased from 0 µg/m³ with neurological disease showing a gradual increase, while the risks for other diseases only increased from 15-20 µg/m³. Effect modification was observed, with children aged 5–9 years at higher risk for all-cause and infectious admissions, and lower-SES regions experiencing greater susceptibility.”

“After adjusting for multiple comparisons using false discovery rate (FDR) method, a 10 µg/m³ increase in LFS PM_{2.5} was associated with increased risks of all-cause [RR: 1.011 (95%CI: 1.009, 1.012)], respiratory [1.019 (1.015, 1.022)], infectious [1.015 (1.010, 1.019)], cardiovascular [1.029 (1.017, 1.041)], neurological [1.028 (1.016, 1.041)], diabetes [1.037 (1.018, 1.057)], cancer [1.015 (1.007, 1.023)], and digestive [1.008 (1.005, 1.012)] hospital admissions. The association with renal disease admissions [1.010 (1.000, 1.020)] was no longer statistically significant after FDR adjustment. (Table 1).”

2. Methods - lack sufficient details in the manuscript.

Are the validation statistics for daily Pm_{2.5} estimates? if note, this should be reported for daily scale as the time-series used daily exposure.

Response:

The validation statistics reported in Xu et al. (Nature, 2023) are based on daily PM_{2.5} estimates, as our study uses daily exposure for the time-series analysis. Xu et al. (Nature,

2023) validated their exposure estimates using tenfold cross-validation, which demonstrated strong performance at a daily scale, with an R^2 of 0.91 and a root mean squared error (RMSE) of $8.47 \mu\text{g}/\text{m}^3$. We have now explicitly stated this in the manuscript to ensure clarity. Please see Lines 313-315.

“We obtained daily LFS $\text{PM}_{2.5}$ concentrations from our previous study, which estimated fire-sourced air pollution at a $0.25^\circ \times 0.25^\circ$ ($\approx 28\text{km} \times 28\text{km}$) spatial resolution using a three-step approach.”

3. Are you using the data reported in the reference no 1? If yes, provide more details about the exposure across your study populations. There is only a few lines about exposure in 3 countries - whether these are for the communities from where health data are obtained is not clear. What about other regions?

Response:

Yes, we used the dataset reported in Xu et al. (Nature, 2023) to obtain daily LFS $\text{PM}_{2.5}$ estimates for all study locations. We acknowledge that our previous description focused only on three countries and lacked details about the other regions. To improve clarity, we have now expanded the Results to include a comprehensive summary of LFS $\text{PM}_{2.5}$ exposure across all seven countries/territories included in our study. These exposure estimates were extracted specifically for the 1012 communities where hospital admission data were collected, ensuring consistency between air pollution exposure and health outcomes. This information has now been incorporated into the Results of the revised manuscript. Please see Lines 75-85.

“Communities in Brazil, Chile, and Thailand experienced higher LFS $\text{PM}_{2.5}$ levels, compared with other regions (Figure 1). The median concentration of LFS $\text{PM}_{2.5}$ was $1.2 \mu\text{g}/\text{m}^3$ [interquartile range (IQR, 25th-75th): $0.4\text{-}3.4 \mu\text{g}/\text{m}^3$], with substantial regional variation (Table S1). Among the study locations, Chile recorded the highest LFS $\text{PM}_{2.5}$ exposure, with a median concentration of $9.5 \mu\text{g}/\text{m}^3$ (IQR: $5.1\text{-}15.5 \mu\text{g}/\text{m}^3$), followed by Brazil ($2.2 \mu\text{g}/\text{m}^3$, IQR: $0.9\text{-}5.7 \mu\text{g}/\text{m}^3$) and Thailand ($1.6 \mu\text{g}/\text{m}^3$, IQR: $0.6\text{-}7.7 \mu\text{g}/\text{m}^3$). Moderate levels were observed in Australia ($1.1 \mu\text{g}/\text{m}^3$, IQR: $0.5\text{-}2.5 \mu\text{g}/\text{m}^3$) and Taiwan ($1.1 \mu\text{g}/\text{m}^3$, IQR: $0.5\text{-}1.9 \mu\text{g}/\text{m}^3$), while New Zealand ($0.7 \mu\text{g}/\text{m}^3$, IQR: $0.3\text{-}1.6 \mu\text{g}/\text{m}^3$) had slightly lower levels. The lowest LFS $\text{PM}_{2.5}$ concentrations were recorded in Canada ($0.3 \mu\text{g}/\text{m}^3$, IQR: $0.1\text{-}1.0 \mu\text{g}/\text{m}^3$). During 2000 to 2019, 67.9 million all-cause [31.4 million (46.3%) girls] hospital admissions among children and adolescents aged 0-19 years were recorded (Table S2).”

4. The method to estimate LFS- $\text{PM}_{2.5}$ is not clear. Are you using the mass fractions of LFS- $\text{Pm}_{2.5}$ from geos-chem and integrating with 0.25 degree downscaled adjusted $\text{Pm}_{2.5}$?

Response:

Yes, we used the mass fractions of fire-sourced $\text{PM}_{2.5}$ from GEOS-Chem. First, we ran two GEOS-Chem simulations (one with fire emissions and one without). The difference gave the fire-sourced $\text{PM}_{2.5}$ at $2.5^\circ \times 2.5^\circ$ resolution. Then, we downscaled both total and fire-sourced $\text{PM}_{2.5}$ to $0.25^\circ \times 0.25^\circ$ resolution using inverse distance weighted interpolation. Next, we calibrated the downscaled total $\text{PM}_{2.5}$ using a random forest model trained on monitoring station data. Finally, we applied the fire-sourced fraction from GEOS-Chem to the bias-corrected total $\text{PM}_{2.5}$ to estimate the final fire-sourced $\text{PM}_{2.5}$

dataset. We have revised the Methods section to clearly describe this process. Please see Lines 313-335.

“We obtained daily LFS PM_{2.5} concentrations from our previous study, which estimated fire-sourced air pollution at a 0.25° × 0.25° (≈28km × 28km) spatial resolution using a three-step approach.

First, fire-sourced and total PM_{2.5} concentrations were estimated using GEOS-Chem by running two simulations: one with fire emissions and one without fire emissions. The difference between these simulations provided the fire-sourced PM_{2.5} concentration at 2.5° × 2.5° resolution. Fire emission data were obtained from the Global Fire Emissions Database (GFED4.1s)³⁹, which captured emissions from six fire sources: boreal forest fires, tropical forest fires, savanna/grassland/shrubland fires, temperate forest fires, peatland fires, and agricultural waste burning. These emissions were derived from satellite retrievals of burned areas and active fire detections. Then fire-sourced and total PM_{2.5} estimates were downscaled from 2.5° × 2.5° to 0.25° × 0.25° using inverse distance weighted (IDW) spatial interpolation⁴⁰.

Second, the downscaled estimates were calibrated using a random forest machine learning model, incorporating ground-based observations from 5,661 monitoring stations across 73 countries. This step reduced bias and improved accuracy in PM_{2.5} estimates.

Third, the fire-sourced fraction of PM_{2.5} from GEOS-Chem was then applied to the bias-corrected total PM_{2.5} estimates to obtain the final fire-sourced PM_{2.5} dataset at a 0.25° × 0.25° resolution. These estimates were validated using spatial tenfold cross-validation, which demonstrated high accuracy for daily total PM_{2.5} estimates, with an R² of 0.91 and a root mean squared error (RMSE) of 8.47 µg/m³.

5. How large are the communities? Since the admission data are from specific hospitals, how large is their catchment area? cities? beyond? Is 0.25 degree spatial scale enough?

Response:

The community sizes in our study range from 391 km² (Australia) to 38,092 km² (Chile), with a median area of 5,384 km². The hospital admission data in our study covered nearly the entire population in most regions, except for Brazil and Chile, where coverage was approximately 70-80%. These community-level hospital admission data provide a finer spatial resolution than city- or country-level datasets, allowing for a more precise assessment of health risks and within-country variations in susceptibility to LFS PM_{2.5}.

To estimate LFS PM_{2.5} exposure, we used a 0.25° × 0.25° spatial resolution, which aligns with the spatial scale of most communities. Rather than using single-point estimates (e.g., from the community centroid), we calculated population-weighted averages of LFS PM_{2.5} across all grid cells within each community. This approach accounts for both spatial variability and the distribution of population exposure, yielding estimates that are representative of actual human exposure patterns. As exposure was averaged across the community area, the difference between 0.25° and finer resolutions (e.g., 0.1°) is unlikely to meaningfully affect the results.

We have clarified community size, hospital catchment areas, and the justification of the 0.25° spatial resolution in the revised Methods section. Please see Lines 291-293, 304-310, 337-343, and Table S1.

“These hospital admission data covered nearly the entire population in most regions, except for Brazil and Chile, where coverage was approximately 70-80% (Table S1).”

“We used community-level hospital admission data because they provided finer spatial resolution than city- or country-level data. The study regions varied in spatial community sizes, with the median area of 5,384.1 km² in total (Table S1). This approach helps capture within-country variations in susceptibility to LFS PM_{2.5} such as categorizing each country based on the SES level of communities, which allows the identification of vulnerable and high-risk communities. Understanding these differences is crucial for targeted interventions and prevention strategies at local levels.”

“We selected a 0.25° × 0.25° spatial resolution for LFS PM_{2.5} exposure assessment, which aligns well with the spatial scale of most communities in our study. Rather than using single-point estimates (e.g., from the community centroid), we calculated population-weighted averages of LFS PM_{2.5} across all grid cells within each community. The calculation method is described in the following section titled “Estimating community-level population-weighted values for exposures”. This approach enhances the representativeness of exposure estimates by accounting for both spatial variability and population distribution.”

6. The statistical analysis section in the main manuscript should briefly discuss the model with details in Method S2. You have adjusted for temperature. Is LFS-PM_{2.5} concentration correlated with temperature? If yes, there could be over-adjustment in the model.

Response:

We have now provided additional details on the statistical model in the main manuscript to enhance clarity. Specifically, we included a description of the cross-basis function for LFS PM_{2.5} (0-14 lag days), which accounts for both nonlinear exposure-response relationships and delayed effects. We also specified the inclusion of potential confounders, such as temperature (0-21 lag days) and relative humidity (0-7 lag days), as well as adjustments for long-term and seasonal trends using natural cubic splines for year and an indicator for the day of the week (DOW).

Temperature is an important confounder in the association between LFS PM_{2.5} and hospital admissions. Previous studies have consistently included temperature in the models to reduce potential residual confounding effects. To assess the correlation between LFS PM_{2.5} and temperature, we conducted a Spearman correlation analysis, which showed a moderate correlation of 0.42. Given this relatively low correlation, adjusting for temperature is unlikely to cause significant over-adjustment but rather helps to control confounding effects.

We have revised the Methods in the manuscript and supplementary file accordingly. Please see Lines 374-378 and Methods S2.

“The model incorporated a cross-basis function for LFS PM_{2.5} (0-14 lag days) to account for both nonlinear exposure-response relationships and delayed effects. We adjusted

potential confounders, including temperature (0-21 lag days) and relative humidity (0-7 lag days). Additionally, natural cubic splines for year and an indicator for the day of the week (DOW) were included to control for long-term and seasonal trends.”

“We adjusted for temperature because it’s a known confounder in the LFS PM_{2.5}-hospital admission associations. Previous studies have consistently included temperature in the models to reduce potential residual confounding effects.”

7. Results are reported for 0 and 1 lag. There are almost negligible effects for lag greater 2 or more days. On the contrary, many time series studies showed effect showing lag for several days. Is it because the LFS generated Pm_{2.5} gets diluted to have any significant effect?

Response:

We acknowledge that while most studies reported the highest effect at lag 0-1 days, consistent with our findings, many studies observed prolonged lag effects of LFS PM_{2.5} exposure on health outcomes. In our study, the largest effects were observed at lag 0-1 days, with minimal associations beyond lag 2 days for most diseases. This pattern aligns with the acute inflammatory and oxidative stress response to LFS PM_{2.5} exposure, which typically peaks within the first 24-48 hours. Additionally, the attenuation of associations at longer lags may result from the dispersion of LFS PM_{2.5} over time, leading to lower exposure levels and reduced direct impact on health.

However, we found that some outcomes, such as digestive disease, showed effects beyond lag 2 days. We provide a more detailed explanation of these prolonged effects in our response to the following Question 8.

We have revised the Discussion to provide a clearer explanation of the observed lag patterns and their potential mechanisms. Please see Lines 194–203.

“The lag patterns observed in our study align with these mechanistic pathways. The observed largest effects at lag 0-1 days suggest an acute inflammation and oxidative stress response to LFS PM_{2.5} exposure, consistent with previous studies¹¹⁻¹⁴. The attenuation of these associations at longer lags may result from both the rapid biological response to acute LFS PM_{2.5} exposure and the dispersion of LFS PM_{2.5} over time, leading to lower exposure levels and reduced direct impact on health³. However, digestive diseases showed a secondary rise in risk around lag 7, potentially due to delayed effects from mucociliary clearance and ingestion of fire-related pollutants²⁶. This prolonged effect highlights the need for clinicians to monitor for delayed-onset symptoms and for healthcare systems to allocate resources accordingly to ensure adequate medical support beyond the immediate exposure period.”

8. For some outcome like digestive, the effect actually increases from lag of 4 days and above - why?

Response:

The delayed risks in digestive hospital admissions after lag 4 days is likely explained by mucociliary clearance and ingestion pathways. Inhaled LFS PM_{2.5} can be transported from the respiratory tract to the gastrointestinal system through mucociliary clearance. Additionally, LFS PM_{2.5} pollutants may be ingested via contaminated food and water, leading to delayed gastrointestinal effects. These pathways may explain why digestive

hospital admissions increase at longer lags compared to other health outcomes. We have revised the Discussion accordingly. Please see Lines 199–203.

“However, digestive diseases showed a secondary rise in risk around lag 7, potentially due to delayed effects from mucociliary clearance and ingestion of fire-related pollutants²⁶. This prolonged effect highlights the need for clinicians to monitor for delayed-onset symptoms and for healthcare systems to allocate resources accordingly to ensure adequate medical support beyond the immediate exposure period.”

9. Fig 3 mentions wildfire-PM2.5. Are you considering only wildfire or all types of fire?

Response:

We consider landscape fires in our study, which include fires in both natural and cultural landscapes (e.g., forests, shrubs, grasslands, pastures, agricultural lands, and peri-urban areas). This encompasses both wildfires (uncontrolled or unplanned fires in wildland vegetation) and human-planned fires (e.g., prescribed burns or agricultural fires). We have revised Figure 3 to ensure consistency in terminology. Please see Figure 3.

10. Xu et al (2023) where LFS-PM2.5 data were generated showed larger uncertainty than for PM2.5. It's not directly possible to evaluate sector-specific PM2.5, but the fact that the effects could have larger errors than reported due to the error in exposure should be acknowledged.

Response:

We acknowledge that LFS PM_{2.5} estimates have greater uncertainty compared to total PM_{2.5} due to the additional modelling steps required to isolate fire-sourced PM_{2.5}. While the exposure data were calibrated and validated, some degree of uncertainty remains. Since we used the population-weighted LFS PM_{2.5} exposure at community level rather than individual-level exposure, this may introduce Berkson error in the exposure assessment. Such error tends to bias the health effect estimates toward the null. Therefore, our findings may underestimate the true health effects of LFS PM_{2.5}. We have added this acknowledgment to the Limitations section in the revised manuscript. Please see Lines 270-277.

“Additionally, the estimation of LFS PM_{2.5} from GEOS-Chem introduces greater uncertainty compared to total PM_{2.5}, as LFS PM_{2.5} relies on model simulations and assumptions about emission sources and chemical transport¹. This may lead to some degree of exposure misclassification. Because exposure estimates were aggregated to the community level using population-weighted averages, such misclassification is likely to be Berkson error, which tends to bias effect estimates toward the null. Therefore, the reported associations may underestimate the true health effects of LFS PM_{2.5}.”

11. For pooled estimates, there is contrast in the observed association - for some diseases some age groups do not show significant association. Any plausible explanation?

Response:

The variation in observed associations across age groups may be explained by differences in exposure levels, physiological susceptibility, and statistical power. Children aged 5-9 years showed higher risks for all-cause and infectious disease admissions, likely due to increased outdoor activity leading to greater exposure, combined with their relatively

immature respiratory and immune systems, making them more susceptible to LFS PM_{2.5} related health effects. In contrast, younger children (0-4 years) may have lower exposure due to spending more time indoors, while older children (10-19 years) may have greater physiological resilience. Additionally, some age-specific associations were not statistically significant, possibly due to limited sample sizes for certain diseases, reducing statistical power to detect significant effects. We have clarified these points in the Discussion. Please see Lines 250-257.

“Significant effect modification by age was observed for all-cause and infectious admissions in our study. Compared with children aged 5-9 years, those aged 0-4 years and 10-19 years showed lower hospital admission risks. Higher outdoor activity levels in children aged 5-9 years likely increased their exposure to LFS PM_{2.5} compared to younger children, while their relatively immature respiratory and immune systems may increase their susceptibility to its health effects compared to older children³⁸. However, these age differences should be interpreted with caution, as some age-specific associations were not statistically significant, possibly due to limited sample sizes for certain diseases.”

Reviewer #3 (Remarks to the Author):

The manuscript titled “Impact of global short-term landscape fire sourced PM2.5 exposure on child cause-specific morbidity: a study in multiple countries and territories” presented a multi-country, time-series study investigating the short-term health impacts of landscape fire-sourced (LFS) PM2.5 on children and adolescents. It analyzed cause-specific hospital admission data from 1012 communities in seven countries/territories (Australia, Brazil, Canada, Chile, New Zealand, Taiwan, and Thailand) from 2000 to 2019. The study stated that short-term exposure to LFS PM2.5 increased hospital admissions for various diseases in children and adolescents. The manuscript is well-structured with robust evidence. I suggest to publish it after moderate revision.

Major comments:

1. “Associations between LFS PM2.5 and hospital admissions were first evaluated at the community level...” Please address the importance to evaluate at community level, what makes it different or more advanced/reasonable compared with other levels (e.g. cities, countries, towns)

Response:

Thanks for your comments. We used community-level hospital admission data to achieve a finer spatial resolution than city- or country-level datasets, allowing for a more precise assessment of health risks. The median community area was 5,384 km², with variations across study regions (Table S1). This approach captures within-country variations in LFS PM_{2.5} exposure and health effects, which may be not evident in larger scale analysis. By analysing at the community level, we can also stratify by socioeconomic status of communities and identify vulnerable and high-risk areas, supporting targeted public health interventions. We have clarified the importance of community-level evaluation in the Methods. Please see Lines 304-310.

“We used community-level hospital admission data because they provided finer spatial resolution than city- or country-level data. The study regions varied in spatial community sizes, with the median area of 5,384.1 km² in total (Table S1). This approach helps capture within-country variations in susceptibility to LFS PM_{2.5} such as categorizing each country based on the SES level of communities, which allows the identification of vulnerable and high-risk communities. Understanding these differences is crucial for targeted interventions and prevention strategies at local levels.”

2. In the introduction section, you did not summarize any other studies (regional- or global-wise) on short-term impacts of LFS PM2.5. Does impacts of LFS PM2.5 mentioned in introduction referred to long-term impacts? I suggest to compare results of other studies on short-term impacts of LFS PM2.5 in introduction or discussion.

Response:

In the Introduction, we initially focused on the general health risks of LFS PM_{2.5}, which may have caused ambiguity regarding the distinction between short-term and long-term effects. To clarify this, we have now explicitly incorporated findings from previous studies on the short-term impacts of LFS PM_{2.5} on hospital admissions, particularly for respiratory diseases, which have been the primary focus of prior research. We also highlight inconsistencies in existing evidence, with some studies reporting significant

associations while others found no effects. Additionally, we have emphasized the limited research on broader health outcomes beyond respiratory diseases in children. These revisions better contextualize our study within the existing literature.

We have also expanded the Discussion to compare our results with previous short-term studies and to explain potential reasons for discrepancies, such as differences in study regions, exposure levels, population characteristics, and healthcare access. Please see Lines 49-57, 159-177.

“Previous studies examining short-term LFS PM_{2.5} exposure in children have primarily focused on respiratory diseases^{8,10}. Some studies conducted in Mozambique¹¹, Brazil¹², California¹³ and Washington State¹⁴ reported positive associations, while others in Colorado¹⁵ and San Diego¹⁶ found no significant effects. A recent review has linked LFS PM_{2.5} to an increased risk of upper respiratory infections, while the evidence remains inconclusive for other health effects, such as asthma and bronchitis¹⁰. However, research on the broader health effects of LFS PM_{2.5} in children is limited, particularly regarding its potential associations with infectious, cardiovascular, neurological, metabolic, digestive, and renal diseases^{8,10}.”

“The relationship between LFS PM_{2.5} and respiratory outcomes aligned with findings from previous studies^{11-14,18}. Specifically, a study from southern Mozambique reported that each 10 µg/m³ increment in cumulative LFS PM_{2.5} for lag 0-1 days was associated with a 15.6% increment in all-cause and a 27.0% increment in respiratory hospital visits¹¹. Another study from Brazil reported a 2.3% increase in all-cause hospital admissions for children aged 5-9 and a 4.9% increase for children aged 0-4¹². Similarly, studies in California¹³ and Washington State¹⁴ also found increased risks of respiratory hospital admissions in children associated with LFS PM_{2.5} exposure for lag 0-1 days (RR: 1.026) and lag 0 day (RR: 1.069), respectively. However, studies in Colorado¹⁵ and San Diego¹⁶ did not observe significant associations. Cardiovascular disease is another major health outcome linked to LFS PM_{2.5} exposure, but most research focused on adults¹⁹. Our study is the first to identify a significant association between LFS PM_{2.5} exposure and higher risk of cardiovascular hospital admissions in children, whereas previous studies in California, Washington, and Colorado did not report significant associations¹³⁻¹⁵. These discrepancies may result from variations in LFS PM_{2.5} concentrations, exposure assessment methods, study populations, and differences in healthcare infrastructure and health-seeking behaviors. Additionally, our study identified new associations between LFS PM_{2.5} with infectious, neurological, diabetes, cancer, and digestive hospital admissions, suggesting a broader impact of LFS PM_{2.5} on child health than previously recognized and reinforcing the need for further research to validate these findings.”

3. I suggest to summarize all the diseases analyzed in results in methods section, and provide reasons why they are selected in the study. Are their associations with LFS PM_{2.5} mentioned in previous studies?

Response:

We have revised the Methods section to clearly summarize all the diseases analysed in our study, including infectious diseases, cancer, diabetes, neurological system disorders, cardiovascular diseases, respiratory diseases, digestive diseases, and renal diseases.

These outcomes were selected to provide a comprehensive assessment of the potential health effects of LFS PM_{2.5}. While previous studies on children have primarily focused on respiratory diseases, mechanistic research suggests that LFS PM_{2.5} can induce systemic inflammation, oxidative stress, and immune dysfunction, potentially affecting multiple organ systems. However, such broader associations have mainly been reported in adults, with limited evidence in children.

Our study fills this gap by examining short-term LFS PM_{2.5} effects across multiple disease categories in children and adolescents. These revisions have been incorporated into the Introduction and Methods section for clarity. Please see Lines 39-44, 59-66, 295-302.

“Inhaled LFS PM_{2.5} can deposit deep into the lungs to trigger respiratory inflammation and oxidative stress, leading to respiratory damage and increased susceptibility to infections⁶. Its toxic components, such as ultrafine particles and heavy metals, can enter the bloodstream to cause systemic inflammation, oxidative stress, and immune dysfunction⁷. These biological pathways suggest that LFS PM_{2.5} may increase risks for multiple organ systems, as widely documented in adults^{4,5}.”

“Most previous studies were conducted in single locations¹¹⁻¹⁷, limiting the ability to compare and synthesize findings across different regions. Although some research spanned multiple countries, they often focused on specific age groups such as 0-5 years or specific health outcomes such as respiratory diseases^{9,18}. This narrow focus might limit the understanding of age-specific vulnerabilities and the broader spectrum of morbidity related to LFS PM_{2.5}. To our knowledge, no previous study has comprehensively evaluated the short-term impacts of LFS PM_{2.5} on a wide range of cause-specific morbidity outcomes in children and adolescents across different regions and populations of the world.”

“In summary, we aggregated and integrated a multi-country dataset of daily age-, sex- and cause-specific hospital admissions for infectious diseases, cancer, diabetes, neurological system disorders, cardiovascular diseases, respiratory diseases, digestive diseases, and renal diseases from 1012 communities across the seven countries/territories, for subsequent analysis. These disease categories were selected due to their biological plausibility and previous associations with LFS PM_{2.5} exposure in adult populations. LFS PM_{2.5} exposure may lead to systemic inflammation, oxidative stress, immune dysfunction, and neurotoxicity, potentially increasing risks for multiple organ systems.”

4. “The highest risks occurred on the current day (lag0) of LFS PM_{2.5} exposure and decreased afterwards, with significantly positive associations on lag0 and lag1 days.” Is the result consistent with previous studies on lag effect as well?

Response:

Our findings align with previous studies showing that the highest risks typically occur on lag 0-1 days, reflecting the acute inflammatory and oxidative stress responses triggered by inhalation of LFS PM_{2.5}. Additionally, the attenuation of associations at longer lags may result from the dispersion of LFS PM_{2.5} over time, leading to lower exposure levels and reduced direct impact on health. Similar lag patterns have been observed in studies on respiratory hospital admissions following LFS PM_{2.5} exposure. However, due to limited evidence on other health outcomes, direct comparisons for other diseases are not available. Notably, we observed a secondary rise in risk for digestive diseases around lag 7, which may be explained by delayed effects from mucociliary clearance and ingestion

of fire-related pollutants via contaminated food and water. We have clarified these findings in the Discussion section. Please see Lines 194-203.

“The lag patterns observed in our study align with these mechanistic pathways. The observed largest effects at lag 0-1 days suggest an acute inflammation and oxidative stress response to LFS PM_{2.5} exposure, consistent with previous studies¹¹⁻¹⁴. The attenuation of these associations at longer lags may result from both the rapid biological response to acute LFS PM_{2.5} exposure and the dispersion of LFS PM_{2.5} over time, leading to lower exposure levels and reduced direct impact on health³. However, digestive diseases showed a secondary rise in risk around lag 7, potentially due to delayed effects from mucociliary clearance and ingestion of fire-related pollutants²⁶. This prolonged effect highlights the need for clinicians to monitor for delayed-onset symptoms and for healthcare systems to allocate resources accordingly to ensure adequate medical support beyond the immediate exposure period.”

5. In discussion section: “We observed that LFS PM_{2.5} was associated with increased risks of all-cause, respiratory, infectious, cardiovascular, neurological, diabetes, cancer, and digestive hospital admissions among children.” Do you mean short-term exposure to LFS PM_{2.5} here? If so, how does it increase chronic diseases as cancer, diabetes?

Response:

Yes, our study focuses on short-term exposure to LFS PM_{2.5} and its association with acute health effects. The observed associations with cancer and diabetes hospital admissions likely reflect exacerbation of pre-existing conditions rather than the new-onset cases.

For paediatric cancer patients (mainly leukemia), short-term exposure to LFS PM_{2.5} may weaken immune function, increase systemic inflammation, and aggravate treatment-related complications, leading to a higher risk of hospitalization. Similarly, for children with Type I diabetes, acute exposure to LFS PM_{2.5} may induce oxidative stress and immune dysfunction, potentially exacerbating their complications and increasing hospitalization risks. We have revised the Discussion to clarify. Please see Lines 216-218.

“..... Especially for pediatric patients with pre-existing diabetes or cancer, short-term LFS PM_{2.5} may compromise their immune function and exacerbate their pollution-induced complications rather than triggering new-onset cases^{24,30}.”

6. In Line 298: “inverse distance weighted spatial interpolation”, I suggest to provide a reference? The fire emission inventory used in GEOS-CHEM model should also be stated here since it is highly related with the study.

Response:

We have made the following revisions:

1) We have added a reference for inverse distance weighted (IDW) spatial interpolation: Lu, G. Y. & Wong, D. W. An adaptive inverse-distance weighting spatial interpolation technique. *Computers & geosciences* 34, 1044-1055 (2008).

2) We have explicitly stated that the fire emission inventory used in GEOS-Chem is the Global Fire Emissions Database (GFED4.1s), which includes emissions from boreal forest fires, tropical forest fires, savanna/grassland/shrubland fires, temperate forest fires, peatland fires, and agricultural waste burning (van der Werf et al., 2017). These details have been added to the revised Methods section. Please see Lines 320-325.

Fire emission data were obtained from the Global Fire Emissions Database (GFED4.1s)³⁹, which captured emissions from six fire sources: boreal forest fires, tropical forest fires, savanna/grassland/shrubland fires, temperate forest fires, peatland fires, and agricultural waste burning. These emissions were derived from satellite retrievals of burned areas and active fire detections. Then fire-sourced and total PM2.5 estimates were downscaled from 2.5° × 2.5° to 0.25° × 0.25° using inverse distance weighted (IDW) spatial interpolation⁴⁰.

7. In Line 309: “Daily relative humidity was calculated from daily temperature and dewpoint temperature.” I believe ERA5 provided relative humidity data as well. Is there a specific reason you need to calculate by yourself?

Response:

ERA5 provides 2m dewpoint temperature, which is a measure of air humidity. Combined with temperature, it can be used to calculate relative humidity. We used this approach to ensure consistency in our meteorological data processing across all study regions. This clarification has been added to the Methods section of the revised manuscript. Please see Lines 346-350.

Daily temperature and 2m dewpoint temperature data at a spatial resolution of 0.25° × 0.25° were derived from the European Centre for Medium-Range Weather Forecasts Reanalysis v5 (ERA5)⁴¹. Since ERA5 provides 2m dewpoint temperature, which is a measure of air humidity, we calculated daily relative humidity using temperature and dewpoint temperature to ensure consistency across study regions⁴².

REVIEWER COMMENTS

Reviewer #1 (Remarks to the Author):

The authors have addressed all my comments, so I recommend the manuscript to be accepted for publication.

Response:

Thanks very much.

Reviewer #3 (Remarks to the Author):

After careful evaluation of the revised manuscript entitled " Impact of global short-term landscape fire sourced PM_{2.5} exposure on child cause-specific morbidity: a study in multiple countries and territories", I find the authors have adequately addressed all review comments. The manuscript now meets the journal's publication standards and is recommended for acceptance. I have just one question:

1. The study explored the short-term landscape fire sourced PM_{2.5} exposure on child cause-specific morbidity. Why the chronic disease (i.e., cancer and diabetes) are selected in this study?

Response:

Thank you for this insightful comment.

Our study aimed to comprehensively assess the short-term health effects of LFS PM_{2.5} exposure on children and adolescents by including a broad range of morbidity outcomes, such as infectious diseases, cancer, diabetes, neurological, cardiovascular, respiratory, digestive, and renal diseases. Chronic conditions like cancer and diabetes were included because short-term exposure to LFS PM_{2.5} may exacerbate pre-existing conditions, leading to increased risk of hospitalisation, rather than initiating new-onset disease.

For example, in paediatric cancer patients (mainly leukemia), short-term LFS PM_{2.5} exposure may impair immune function and increase systemic inflammation, potentially aggravating treatment-related complications. Similarly, in children with Type I diabetes, acute exposure may induce oxidative stress and immune dysregulation, increasing the likelihood of acute complications that necessitate hospital care.

These explanations are detailed in the Discussion, Lines 213–226:

“The exposure-response relationships observed further supported these mechanistic pathways.In contrast, cardiovascular (mainly peripheral vascular diseases in children of this study), diabetes (mainly Type I diabetes), cancer (mainly leukemia), digestive (mainly acute pancreatitis), and renal (mainly renal tubulointerstitial diseases) risks increased only at higher concentrations, around 15-20 µg/m³, suggesting more sustained or higher exposure is needed to trigger systemic effects through oxidative stress and inflammation. Epecially for pediatric patients with pre-existing diabetes or cancer, short-term LFS PM_{2.5} may compromise their immune function and exacerbate their pollution-induced complications rather than triggering new-onset cases ^{24,30}.”

2. fire sourced PM_{2.5} exposure was estimated using GFED4.1s, some small fire (i.e., agricultural biomass burning) might not be detected, which might underestimate the result. Please discuss this issue.

Response:

Thanks for your comments.

We agree that although GFED4.1s is a widely used and comprehensive dataset for estimating global fire emissions, it may under-detect small fires, such as agricultural

biomass burning, due to limitations in satellite resolution and interference from cloud cover or vegetation. This underestimation could lead to non-differential exposure misclassification, which would likely bias the results toward the null and suggest that our reported associations may underestimate the true health risks of LFS PM_{2.5} exposure.

We have now included this limitation in the revised discussion, Lines 280-290.

“Additionally, the estimation of LFS PM_{2.5} from GEOS-Chem introduces greater uncertainty compared to total PM_{2.5}, as LFS PM_{2.5} relies on model simulations and assumptions about emission sources and chemical transport ¹. Fire emission inputs were based on the Global Fire Emissions Database (GFED4.1s), a widely used and comprehensive dataset. However, it may under-detect small fires, such as agricultural biomass burning, due to limitations in satellite resolution and interference from cloud cover or vegetation. This may lead to some degree of exposure misclassification. Because exposure estimates were aggregated to the community level using population-weighted averages, such misclassification is likely to be Berkson error, which tends to bias effect estimates toward the null. Therefore, the reported associations may underestimate the true health effects of LFS PM_{2.5}.”

3.L439: “changing the df for meteorological variables: temperature from 4 to 3 and 5, RH from 3 to 4 and 5. 3)”. Should the later reference to 5.3 be corrected to df=5?”

Response:

Thank you for pointing this out.

This was not intended to be “5.3” but rather part of the numbering of the sensitivity analyses. To avoid any potential confusion, we have revised the punctuation to improve clarity. The sentence now reads as follows (Line 408-410):

“2) changing the df for meteorological variables: temperature from 4 to 3 and 5, relative humidity from 3 to 4 and 5; 3) changing the df for time trends from 7 to 8 and 9.”

Reviewer #4 (Remarks to the Author):

1. Why these countries? Which areas are at higher risk of LFS PM_{2.5}? Lower risk? This could be included in the introduction.

Response:

Thank you for this thoughtful comment.

We selected the seven countries/territories (Brazil, Chile, Australia, Canada, Thailand, New Zealand, and Taiwan) to reflect a wide range of fire activity patterns, geographic and socioeconomic contexts, and the availability of high-quality daily hospital admission data. Brazil, Chile, Australia, Canada, and Thailand are fire-prone regions with frequent wildfires or agricultural burning. In contrast, New Zealand and Taiwan are less fire-prone but can be affected by long-range smoke transport. Including both fire-prone and non-fire-prone regions enabled us to assess the health effects of both local and transported LFS PM_{2.5} exposure.

We have revised the Introduction to highlight this geographic variation and to clarify the rationale for including both types of regions. Please see Lines 37-42, 75-78.

“LFS PM_{2.5} exposure shows substantial geographic variation. High levels are typically observed in fire-prone regions such as South America, Southeast Asia, and northern Australia, while non-fire-prone areas may also experience considerable exposure due to long-range smoke transport¹. During transport, LFS PM_{2.5} can become more toxic through chemical transformation, potentially increasing its health impacts^{1,3}.”

“Specifically, our dataset includes both fire-prone regions (Brazil, Chile, Australia, Canada, and Thailand) and non-fire-prone regions (New Zealand and Taiwan), allowing us to examine the health effects of both local and transported LFS PM_{2.5} exposure.”

2. Can you explain the cardiovascular hospital admissions in children? This is not necessarily expected in this age group and warrants further explanation.

Response:

Thank you for this insightful comment.

Our study aimed to comprehensively assess short-term health effects of LFS PM_{2.5} exposure in children by including a broad range of morbidity outcomes. Although cardiovascular hospital admissions are relatively uncommon in children, short-term exposure to LFS PM_{2.5} may exacerbate pre-existing cardiovascular conditions, such as congenital heart disease or myocarditis, leading to hospitalisation.

As noted in the Lines 187–192, toxic components of LFS PM_{2.5} can enter the bloodstream after inhalation, inducing systemic inflammation and oxidative stress, which may affect the cardiovascular system. As described in Lines 219–224, the exposure-response relationship for cardiovascular diseases became evident only at higher concentrations of LFS PM_{2.5} (around 15-20 µg/m³). This contrasts with the patterns for respiratory and infectious outcomes, which showed elevated risks even at lower exposure levels. The higher threshold for cardiovascular effects may reflect the need for more sustained or intense exposure to trigger systemic inflammation in children

To clarify this point, we have added the following sentence to the Discussion. Please see Lines 226–229.

“Although cardiovascular hospital admissions are relatively uncommon in children, short-term exposure to LFS PM_{2.5} may worsen pre-existing conditions such as congenital heart disease or myocarditis and lead to hospitalisation.”

3. The last sentence of the first paragraph of the methods needs references (lines 346-348).

Response:

Thanks for your comment.

We have now added appropriate references to support the statement regarding the potential biological mechanisms linking LFS PM_{2.5} exposure to health effects. Please see Lines 313-315 and references 7, 20-24.

“LFS PM_{2.5} exposure may lead to systemic inflammation, oxidative stress, immune dysfunction, and neurotoxicity, potentially increasing risks for multiple organ system^{7, 20-24}.”

Reference

7 Cascio, W. E. Wildland fire smoke and human health. *Sci Total Environ* 624, 586-595, doi:10.1016/j.scitotenv.2017.12.086 (2018).

20 Po, J. Y., FitzGerald, J. M. & Carlsten, C. Respiratory disease associated with solid biomass fuel exposure in rural women and children: systematic review and meta-analysis. *Thorax* 66, 232-239, doi:10.1136/thx.2010.147884 (2011).

21 Roscioli, E. et al. Airway epithelial cells exposed to wildfire smoke extract exhibit dysregulated autophagy and barrier dysfunction consistent with COPD. *Respiratory Research* 19, 234, doi:10.1186/s12931-018-0945-2 (2018).

22 Sampath, V., Aguilera, J., Prunicki, M. & Nadeau, K. C. Mechanisms of climate change and related air pollution on the immune system leading to allergic disease and asthma. *Seminars in Immunology* 67, 101765, doi:<https://doi.org/10.1016/j.smim.2023.101765> (2023).

23 Chen, H., Samet, J. M., Bromberg, P. A. & Tong, H. Cardiovascular health impacts of wildfire smoke exposure. *Particle and fibre toxicology* 18, 2, doi:10.1186/s12989-020-00394-8 (2021).

24 Lei, Y., Lei, T.-H., Lu, C., Zhang, X. & Wang, F. Wildfire Smoke: Health Effects, Mechanisms, and Mitigation. *Environmental science & technology* 58, 21097-21119, doi:10.1021/acs.est.4c06653 (2024).

4. For the description of the LFS PM_{2.5} estimation, please provide a reference to the previous study (Lines 359-361).

Response:

Thanks for your comment.

We have now added appropriate references our previous study that describes the methodology for estimating LFS PM_{2.5} concentrations. Please see Lines 326-328 and reference 1.

“We obtained daily LFS PM_{2.5} concentrations from our previous study, which estimated fire-sourced air pollution at a 0.25° × 0.25° (≈28km × 28km) spatial resolution using a three-step approach¹.”

Reference

1 Xu, R. et al. Global population exposure to landscape fire air pollution from 2000 to 2019. Nature 621, 521-529, doi:10.1038/s41586-023-06398-6 (2023).

5. A recommendation is to also include the median population size of the communities here as well. (Line 352)

Response:

Thank you for the suggestion.

We have added the median population size to the sentence describing the community characteristics. Please see Lines 318–319:

“The study regions varied in spatial community sizes, with a median area of 5,384.1 km² and a median population of 126,708 (Table S1).”

6. How was the false discovery rate adjustment conducted?

Response:

Thank you for your comment.

To control for Type I error due to multiple testing, we applied a false discovery rate (FDR) adjustment using the Benjamini-Hochberg (BH) method. This was implemented in R using the p.adjust function with method = "fdr".

We have now clarified this in the Methods section. Please see Lines 397-399.

“We used the false discovery rate (FDR) adjustment to control for Type I error due to multiple testing. The adjustment was implemented using the Benjamini-Hochberg (BH) method via the p.adjust function (method = "fdr") in R”

7. Is there a typo for the RH in line 441?

Response:

Thank you for your comment.

“RH” refers to relative humidity. To avoid any confusion, we have removed the abbreviation and now spell out “relative humidity” in the revised sentence. Please see Line 408-409.

“2) changing the df for meteorological variables: temperature from 4 to 3 and 5, relative humidity from 3 to 4 and 5;”